# A human isogenic iPSC-derived cell line panel identifies major regulators of aberrant astrocyte proliferation in Down syndrome

Keiji Kawatani[1], Toshihiko Nambara[1], Nobutoshi Nawa [1], Hidetaka Yoshimatsu[1], Haruna Kusakabe[1], Katsuya Hirata [1,2], Akira Tanave[3], Kenta Sumiyama [3], Kimihiko Banno[1,4], Hidetoshi Taniguchi [1], Hitomi Arahori[1], Keiichi Ozono[1] & Yasuji Kitabatake [1✉]

Astrocytes exert adverse effects on the brains of individuals with Down syndrome (DS). Although a neurogenic-to-gliogenic shift in the fate-specification step has been reported, the mechanisms and key regulators underlying the accelerated proliferation of astrocyte precursor cells (APCs) in DS remain elusive. Here, we established a human isogenic cell line panel based on DS-specific induced pluripotent stem cells, the *XIST*-mediated transcriptional silencing system in trisomic chromosome 21, and genome/chromosome-editing technologies to eliminate phenotypic fluctuations caused by genetic variation. The transcriptional responses of genes observed upon *XIST* induction and/or downregulation are not uniform, and only a small subset of genes show a characteristic expression pattern, which is consistent with the proliferative phenotypes of DS APCs. Comparative analysis and experimental verification using gene modification reveal dose-dependent proliferation-promoting activity of *DYRK1A* and *PIGP* on DS APCs. Our collection of human isogenic cell lines provides a comprehensive set of cellular models for further DS investigations.

[1] Department of Pediatrics, Graduate School of Medicine, Osaka University, Suita, Osaka, Japan. [2] Department of Neonatal Medicine, Osaka Women's and Children's Hospital, Izumi, Osaka, Japan. [3] Laboratory for Mouse Genetic Engineering, RIKEN Center for Biosystems Dynamics Research, Suita, Osaka, Japan. [4] Department of Physiology II, Nara Medical University, Kashihara, Nara, Japan. ✉email: ykitaba@ped.med.osaka-u.ac.jp

Cellular and organismal homoeostasis relies on a precise and delicate balance among various gene pathways[1]. To maintain this intricate equilibrium, protein production is strictly controlled by chromatin structure, transcriptional regulation, or posttranscriptional modification[2]. Alteration of gene dosage is one of the leading causes of a perturbed transcriptional network, leading to severe impacts on biological processes underlying cellular functions. Down syndrome (DS; trisomy 21) is the most common form of chromosomal aberration, which results from an extra copy of human chromosome 21 (refs. [3,4]). All individuals with DS exhibit various types of clinical features, including intellectual disability and cognitive deficits[5]. Although the healthy human brain contains almost equal numbers of neuronal and glial cells[6], studies with DS brains showed a significantly reduced number of neurons and nearly twice as many astrocytes compared with that of age-matched controls[7–9]. In general, astrocytes are mainly generated by two different processes: fate specification from multipotent progenitor cells and the local proliferation of lineage-specific (i.e. differentiated) astrocyte precursor cells (APCs)[10]. A neurogenic-to-gliogenic shift in the timing of fate specification from radial glia or neural progenitor cells (NPCs) in DS has been reported, and several genes have been identified as causative regulators[11–15]. However, a major source of astrocytes in the postnatal cortex is attributable to a local division of differentiated APCs, rather than differentiation from progenitor cells, and approximately half of all mature cortical astrocytes are produced by APC proliferation[16]. Because these increased astrocytes can interact negatively with neurons during neuronal maturation, synapse formation, and the release of factors that promote neuronal apoptosis[17–19], elucidating whether and how dysregulated APC proliferation is involved in DS pathophysiology is essential.

Generating systematic and precise disease models and identifying the responsible genes are crucial for investigating underlying disease mechanisms. Despite the rapid development of disease-specific, induced pluripotent stem cells (iPSCs) and genome-editing technologies, the complexity of transcriptional dynamics affected by the extra copy of chromosome 21 and fluctuations of gene-expression profiles across individuals or cell lines hinders the identification of key regulators. An innovative and excellent model system for studying the biology of DS has been developed by integrating the human X-inactive specific transcript (*XIST*) gene into DS-specific iPSCs[20]. In one such cell line, *XIST* was inserted into a copy of chromosome 21 in trisomy 21 iPSCs (Tri21 iPSCs), and a long noncoding RNA induced a series of chromatin modifications that stably silenced gene transcription across the whole chromosome in *cis*. Chromosome silencing occurred even in differentiated cells, and various pathologies observed in DS (including proliferative defects, impaired neural differentiation, and haematopoietic abnormalities) were successfully reversed by transcriptional inactivation of the supernumerary chromosome[21,22]. Using this genome-silencing technology, where *XIST* RNA expression is regulated by the tetracycline-inducible system, enables researchers to investigate the correlation between gene-expression changes and cellular phenotypes in DS, without limitations caused by transcriptional heterogeneity and differences among cell lines.

To eliminate biological 'noise', which can result from genetic variability, we established an isogenic iPSC panel where all cell lines share a single genetic background by combining DS-specific iPSCs, *XIST*-induced chromosome silencing, and genome/chromosome-editing technologies (Fig. 1). These cell lines were subjected to astrocytic differentiation, and comparative analysis between their gene-expression profiles and proliferative phenotypes (with a common genetic background) was performed. Once *XIST*-induced silencing was stabilised, the transcriptional levels of most genes were continuously suppressed after the removal of doxycycline (Dox). However, the enhanced proliferative phenotype of APCs in DS, which was suppressed by chromosome silencing, returned to aberrantly accelerated conditions by Dox removal. Careful analysis of this discrepancy between transcriptional and phenotypic responses enabled us to narrow down the causative genes responsible for APC overproliferation. We further established various types of systematically designed partial trisomy 21 iPSCs (Partial-Tri21 iPSCs), leading to the identification of two responsible genes, namely dual-specificity tyrosine-phosphorylation-regulated kinase 1A (*DYRK1A*) and phosphatidylinositol glycan anchor biosynthesis, class P (*PIGP*).

## Results

### Generation of an isogenic iPSC panel for disease modelling of trisomy 21.

We previously generated a patient-derived Tri21 iPSC line that contains one paternal copy and two maternal copies of chromosome 21 (ref. [23]). Using this DS-specific iPSC line, we further generated a corrected-disomy 21 iPSC line (cDi21 iPSC), in which a single copy of chromosome 21 was artificially removed from a Tri21 iPSC line[24], and a partial trisomy 21 iPSC line (Partial-Tri21 iPSC) in which a 4-megabase (Mb) region corresponding to a 'Down syndrome critical region' was selectively deleted only from the paternal chromosome 21 in Tri21 iPSCs[23] (Fig. 1 and Supplementary Table 1). We further generated an *XIST*-mediated chromosome 21-silencing system (Supplementary Fig. 1)[20]; the resultant iPSC clone (*XIST*-Tri21 iPSC) exhibited a typical morphology, expression of pluripotent markers, and a trisomy 21 karyotype (Supplementary Fig. 2a, b). Chromosome 21 in *XIST*-Tri21 iPSCs contained one paternal chromosome and two maternal chromosomes, which is consistent with that in the original iPSC (Supplementary Fig. 2c). Administration of Dox for 3 weeks (D+) successfully induced *XIST* RNA expression and accumulation of H3K27me3, a hallmark of heterochromatin, which led to transcriptional silencing of genes on chromosome 21 (Supplementary Fig. 3a–f). Transduction of reverse tetracycline transactivator (rtTA) into NPCs differentiated from *XIST*-Tri21 iPSCs (*XIST*-Tri21 NPCs) using a *piggyBac* (PB) transposon vector and a hyperactive PB transposase[25] significantly elevated *XIST* RNA expression and H3K27me3 signals after 5 days of Dox administration (Supplementary Fig. 4a–f). Single-nucleotide polymorphism (SNP) analysis of mRNAs extracted from *XIST*-Tri21 APCs (with or without Dox treatment) showed that an SNP in the *ETS2* gene (rs457705) derived from the paternal (P) allele (T→G) was conserved after *XIST*-mediated chromosome silencing. This indicates that the *XIST* cDNA cassette was inserted in one of the two maternal copies (M1 and M2) of chromosome 21 (Supplementary Fig. 5; the *XIST*-inserted maternal chromosome is hereafter referred to as M2).

### *XIST*-mediated chromosome silencing affects the overproliferative phenotypes of DS APCs.

Among the various pathological features of DS, we focused on the aberrantly increased astrocyte population in the brains of individuals with DS. *XIST*-Tri21 NPCs transfected with an rtTA-expression vector were differentiated into the astrocyte lineage (Fig. 2a). Glial fibrillary acidic protein (GFAP) and S100β (typical astrocytic markers) were detected in over 90% of the differentiated cells (Fig. 2b). Moreover, nearly all differentiated cells were positive for CD44 or vimentin (astrocyte-restricted precursor cell markers)[11,26], but negative for SOX1 (an early marker for neural stem cells), indicating that the cells differentiated to APCs (*XIST*-Tri21 APCs). *XIST*-Tri21 APCs exhibited sufficient rtTA expression and Dox-dependent induction of *XIST* RNA

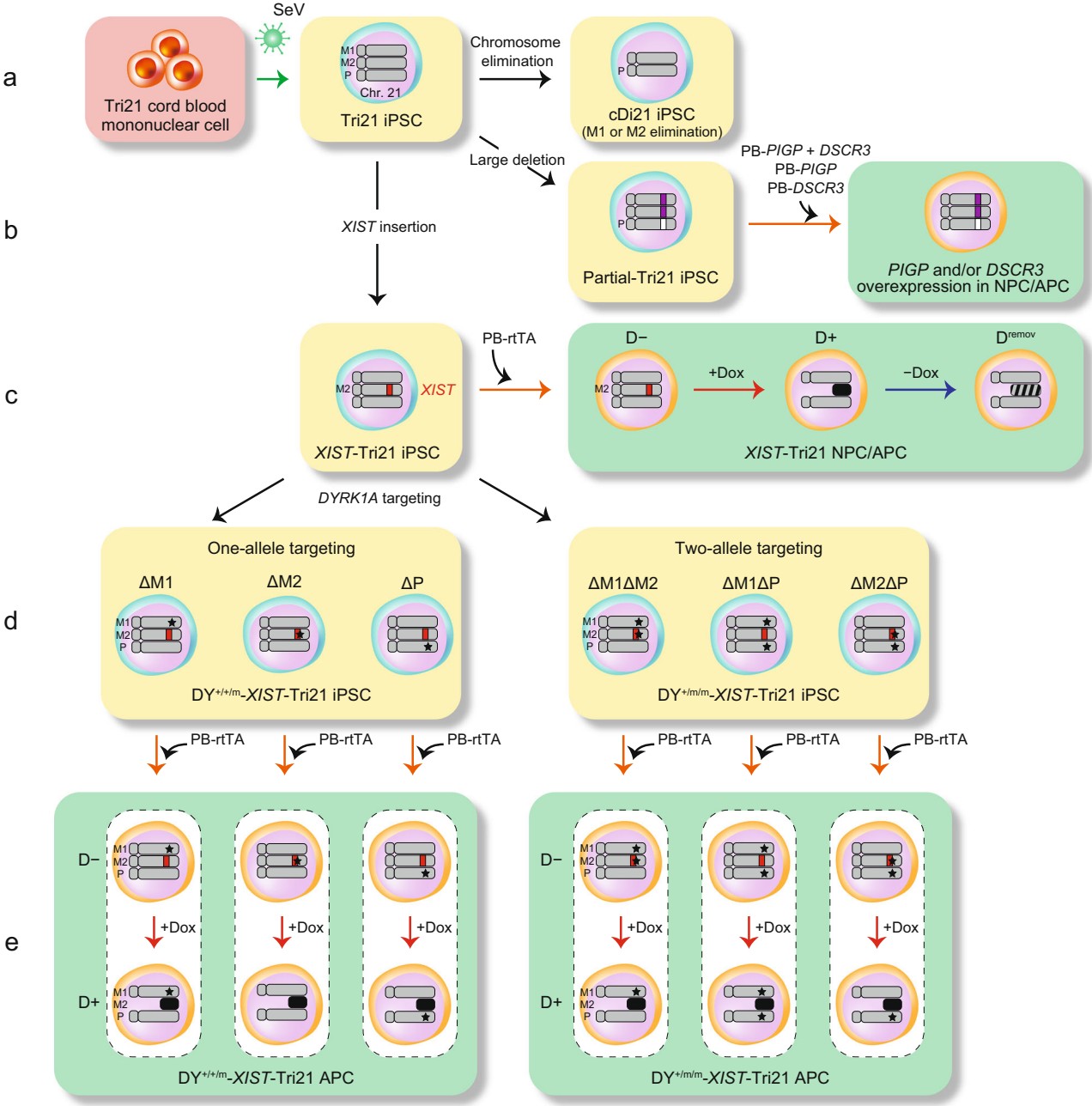

**Fig. 1 Generation of an isogenic iPSC panel for disease modelling of DS.** Schematic overview depicting the strategy used to generate isogenic iPSC lines by combining *XIST*-induced chromosome silencing and genome/chromosome-editing technologies. **a** Cord blood mononuclear cells were derived from a male patient with DS. Then, a DS-specific iPSC line (Tri21) was directly established using Sendai virus. A corrected-disomy cell line (cDi21) was generated by eliminating a single copy of chromosome 21 from a Tri21 iPSC line. **b** A 4-Mb region corresponding to a 'Down syndrome critical region' was selectively deleted only from the paternal chromosome 21 in Tri21 iPSCs to generate Partial-Tri21 iPSC. NPCs differentiated from Partial-Tri21 iPSCs were transfected with *piggyBac* (PB) transposon vectors containing human *PIGP* and/or *DSCR3* cDNA under the regulation of the Tet-inducible system. NPCs were differentiated into APCs for further analysis. **c** Dox-inducible human *XIST* cDNA was inserted into one copy of chromosome 21 in Tri21 iPSCs (*XIST*-Tri21 iPSC) for chromosome silencing. rtTA was additionally introduced into NPCs differentiated from *XIST*-Tri21 iPSCs, using a PB vector. Transcriptional and phenotypic analysis was conducted in Dox-untreated (D−) and Dox-treated (D+) cells, and in NPCs or APCs at 3 weeks after Dox removal (D$^{remov}$). The location of the *XIST* transgene is indicated by the red rectangle. The black and striped chromosomes indicate a silenced and a reverted chromosome 21, respectively. **d** The *DYRK1A* gene was targeted in one or two chromosomes in the *XIST*-Tri21 iPSC line to generate DY$^{+/+/m}$- and DY$^{+/m/m}$-*XIST*-Tri21 iPSCs, respectively. Three types of genome-edited cell lines were generated in each single- or double-targeted clone, based on the parental origin of the three chromosomes. Black stars indicate the *DYRK1A* target sites. Two maternal copies and one paternal copy of chromosome 21 are indicated as M1/M2 and P, respectively. **e** *DYRK1A*-targeted iPSC lines were differentiated to APCs and subjected to phenotypic analysis. Light yellow panels and light green panels indicate iPSC lines and NPC/APC lines, respectively.

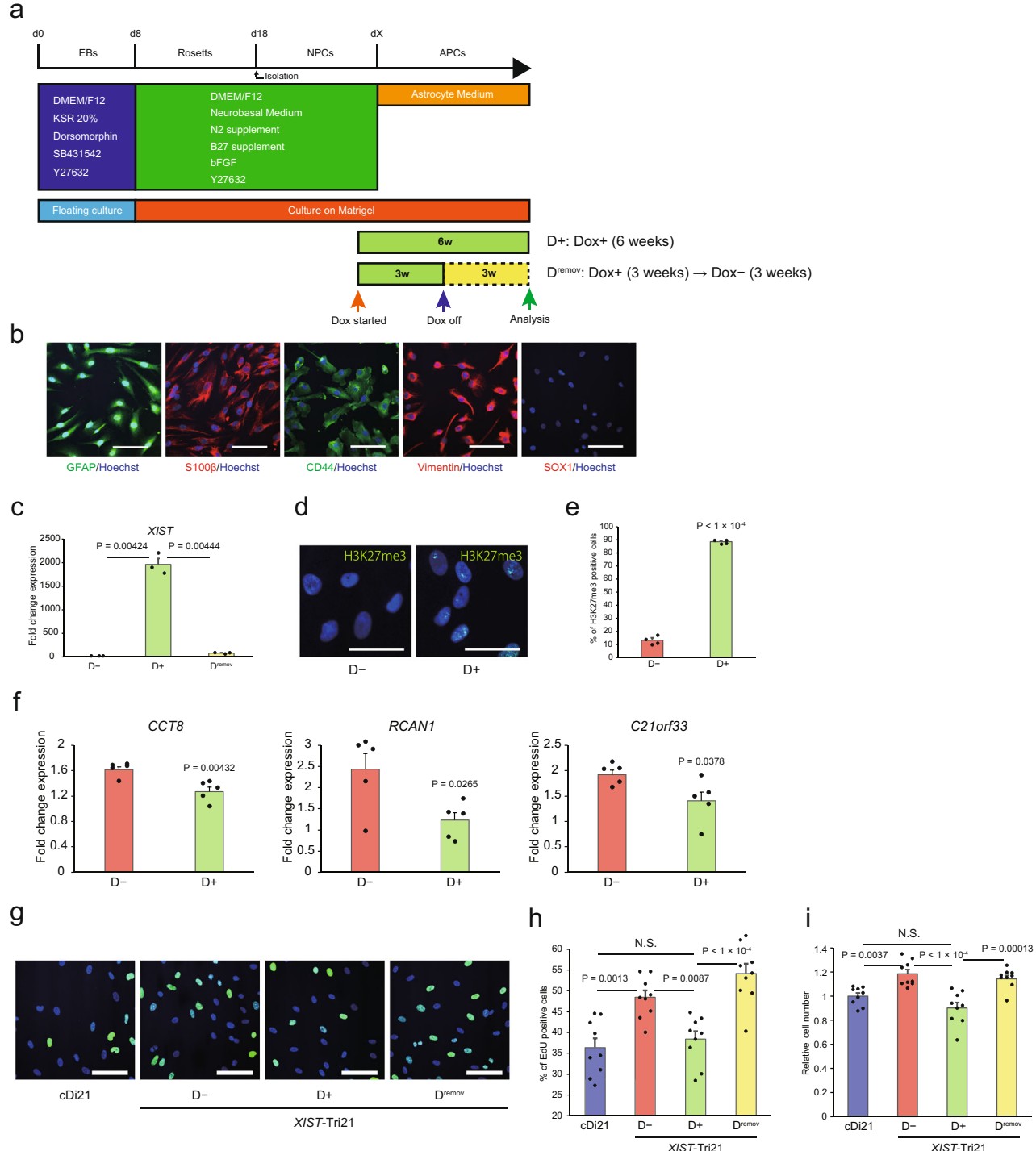

**Fig. 2 *XIST*-mediated chromosome silencing affects the overproliferative phenotype of DS APCs. a** Schematic depiction of the differentiation protocol for NPCs and APCs. EBs embryoid bodies. **b** Immunocytochemistry of *XIST*-Tri21 APCs using GFAP and S100β (astrocyte markers), CD44 and vimentin (APC markers), and SOX1 (NPC marker). Nuclei were stained with Hoechst 33342. Scale bars: 100 μm. **c** Relative expression levels of *XIST* RNA in *XIST*-Tri21 APCs. Expression was normalised to that of D− APCs ($n = 3$ experiments per cell line). **d** Immunocytochemistry of *XIST*-Tri21 NPCs using an H3K27me3-specific antibody. Nuclei were stained with Hoechst 33342. Scale bars: 50 μm. **e** Percentage of H3K27me3-positive cells in *XIST*-Tri21 APCs ($n = 4$ experiments per cell line). **f** Relative expression levels of genes on chromosome 21 in *XIST*-Tri21 APCs. Expression levels were normalised to those of cDi21 cells ($n = 5$ experiments per cell line). **g** EdU assay in *XIST*-Tri21 APCs. EdU-positive cells were stained with an Alexa Fluor 488-conjugated antibody (light green). Nuclei were stained with Hoechst 33342. Scale bars: 100 μm. **h** Percentage of EdU-positive APCs ($n = 3$ independent differentiation experiments per cell line, performed in triplicate). **i** Relative numbers of APCs 1 day after seeding. Cell numbers were normalised to those of cDi21 lines ($n = 3$ independent differentiation experiments per cell line, performed in triplicate). Data were obtained from three lines (**c**, **e**, **f**, **h**, **i**). Error bars represent the SEM. Data were analysed by Student's *t*-test (**e**, **f**), Welch's two-sample *t*-test (**c**), or one-way ANOVA with Bonferroni's correction (**h**, **i**). NS not significant ($P > 0.05$). EdU 5-ethynyl-2′-deoxyuridine.

expression and H3K27me3 marks (Fig. 2c–e and Supplementary Fig. 6a). The expression levels of genes on chromosome 21 in Dox-treated XIST-Tri21 APCs were lower than in Dox-untreated cells, suggesting that XIST-mediated chromosome silencing was successfully induced in APCs (Fig. 2f).

Cell-proliferation assays performed using 5-ethynyl-2′-deoxyuridine (EdU) in Tri21 APCs showed a higher proliferation rate than that of isogenic euploid APCs (cDi21 APCs). Dox administration (D+) did not affect the basal proliferative ability of simple Tri21 APCs (i.e. those without XIST cDNA) but significantly decreased the proliferation rate of XIST-Tri21 APCs to a level similar to that of cDi21 cells (Fig. 2g–i and Supplementary Fig. 6b, c). Furthermore, removing Dox (D^remov) for 3 weeks after the initial 3-week Dox treatment increased the proliferation rate of XIST-Tri21 APCs to that of Dox-untreated cells (Fig. 2g–i). This proliferative change was accompanied by reduced XIST expression (Fig. 2c), suggesting that the expression of genes responsible for APC overexpression is suppressed by chromosome silencing, which was reversed by Dox removal. Meanwhile, astrocytic markers were stably expressed throughout these treatments (Supplementary Table 2).

**XIST-mediated chromosome silencing is generally maintained for at least 3 weeks after Dox removal.** Previous reports showed that inducible expression of murine Xist initially causes reversible chromosome inactivation in undifferentiated cells followed by irreversible inactivation after differentiation[27], whereas forced expression of human XIST cDNA in somatic cells results in reversible silencing[28]. To assess how induction and depletion of ectopic XIST affect transcriptional dynamics in human differentiated cells, gene expression and H3K27me3 histone-methylation profiles were analysed in four cell lines, i.e. cDi21 APCs or XIST-Tri21 APCs, with (D+)/without (D−) Dox, and 3 weeks after Dox removal (D^remov).

Overall, gene expression from chromosome 21 was higher in D− APCs than in cDi21 APCs, probably due to the gene-dosage effects of trisomy 21. Mean expression levels of chromosome 21 in XIST-Tri21 APCs were suppressed to similar levels as those in cDi21 cells after Dox treatment (relative ratio to D−, D+, and D^remov lines was 1.319, 1.026, and 1.079, respectively; Fig. 3a). Consistent with previous data indicating that XIST expression is essential for initiating chromosome silencing but is not required for chromosome maintenance[20,27,29], Dox treatment effectively suppressed the expression of genes located in chromosome 21, which was overall maintained for 3 weeks after Dox was removed (Fig. 3a, Supplementary Figs. 7 and 8 and Supplementary Data 1). This observation was supported by hierarchical-cluster analysis of 178 genes with positive read counts on chromosome 21, demonstrating relative similarities in the transcriptional profiles of D+ and D^remov APC lines compared with the D− line (Fig. 3b and Supplementary Data 1 and 2). H3K27me3 marks were enriched along chromosome 21 in D+ APCs (Fig. 3c), whereas the levels were lower in D^remov APCs but substantially more predominant than in D− cells. A clear, negative correlation between H3K27me3 enrichment and reduced expression was observed among genes on chromosome 21 in D+ and D^remov APCs (Fig. 3c and d and Supplementary Fig. 9). These results indicate that forced expression of human XIST effectively inactivates chromosome 21, which can be generally maintained for at least 3 weeks after Dox removal.

**The 4-Mb region on chromosome 21 is critical for aberrant APC proliferation in DS.** Although the results of RNA sequencing (RNA-seq) and chromatin immunoprecipitation-sequencing (ChIP-seq) indicated that the transcriptional and epigenetic

changes induced by XIST-mediated chromosome silencing were preserved for at least 3 weeks after Dox removal, the proliferation rate of APC was reversed from the suppressed condition in D+ cells to the acceleration in D^remov cells (Fig. 2g–i). To explore the underlying mechanisms that can explain this difference, we focused on a shape change in the violin plot for D^remov cells. In the violin plot, other cell lines showed uniform distribution while D^remov cells showed a bimodal distribution, as evidenced by the occurrence of an additional hump in the plot (Fig. 3a, arrow), suggesting the existence of a small subgroup of genes whose expression was rapidly restored after the removal of Dox-induced silencing. We speculated that this phenomenon was simulated in component 1 of our principal component analysis (PCA), in which the values of D^remov lines moved relatively close to those of the D− lines while drifting away from those of the D+ and cDi21 lines (Fig. 4a).

To identify causative genes among the list of genes in component 1 (Supplementary Table 3), we narrowed down the candidate genes (Fig. 4b). Among the 517 genes located on chromosome 21, nearly one-third (178 genes) showed positive read counts in at least one of the APC lines and 56 genes showed ≥1.5-fold increased expression compared with the corresponding expression levels in cDi21 APCs. The expression levels of 21 coding genes were significantly decreased in D+ cells, suggesting that these genes are involved in the gene dosage-dependent phenotypic alteration found in XIST-Tri21 APCs.

With the Partial-Tri21 iPSC line[23] (Fig. 4c), the deletion of the 4-Mb region from one chromosome 21 led to a significant decrease in the proliferation rate of APCs to the same level as that of cDi21 APCs, indicating that this segment is responsible for the aberrant proliferative properties of DS APCs (Fig. 4d, e). Among the 178 genes with the positive expression on chromosome 21, 24 genes located in this 4-Mb region were listed in order of the values of PCA component 1 (Table 1); three genes (DYRK1A, DSCR3, and HLCS) were ranked in the top 10, and four genes (MORC3, PIGP, CBR3, and SETD4) were in the top 50 (25, 31, 36, and 44, respectively). Of these seven genes, MORC3 and SETD4 were excluded for their low expression in D− (<1.5-fold) and CBR3 was excluded for its low expression in D^remov. Consistent with the PCA results, RNA-seq data demonstrated that expression levels of the remaining four genes (DYRK1A, DSCR3, HLCS, and PIGP) reverted after Dox removal (Table 1). The gene-expression alterations were validated by quantitative reverse transcriptase-polymerase chain reaction (qRT-PCR) for the DYRK1A, PIGP, and DSCR3 genes but not for the HLCS gene (Fig. 4b, f). Significantly increased expression and elimination of H3K27me3 deposition were observed for these genes in D+ and D^remov cells, respectively (Fig. 4g and Supplementary Fig. 10a, b). Thus, DYRK1A, PIGP, and DSCR3 were identified as candidate causative genes of aberrant APC proliferation in DS.

**DYRK1A potently regulates APC proliferation in a gene dosage-dependent manner.** Among the 24 genes in the 4-Mb region, DYRK1A was the most likely candidate gene for APC proliferation in the PCA component 1 list (Table 1). However, it is well established that DYRK1A exerts dose-dependent antiproliferative activity in several cell types, especially in neural precursor cells[30,31]. To investigate whether DYRK1A conversely promotes APC proliferation and whether it acts independently in DS pathophysiology or cooperates with other molecules synergistically, we prepared several genome-edited Tri21 iPSC lines, in which DYRK1A was targeted in one or two chromosomes in the XIST-Tri21 iPSC line (DY^+/+/m_ and DY^+/m/m_-XIST-Tri21 iPSC, respectively; Fig. 5a). The targeting cassette was inserted into the DYRK1A locus in XIST-Tri21 iPSCs using clustered regularly interspaced short palindromic repeats

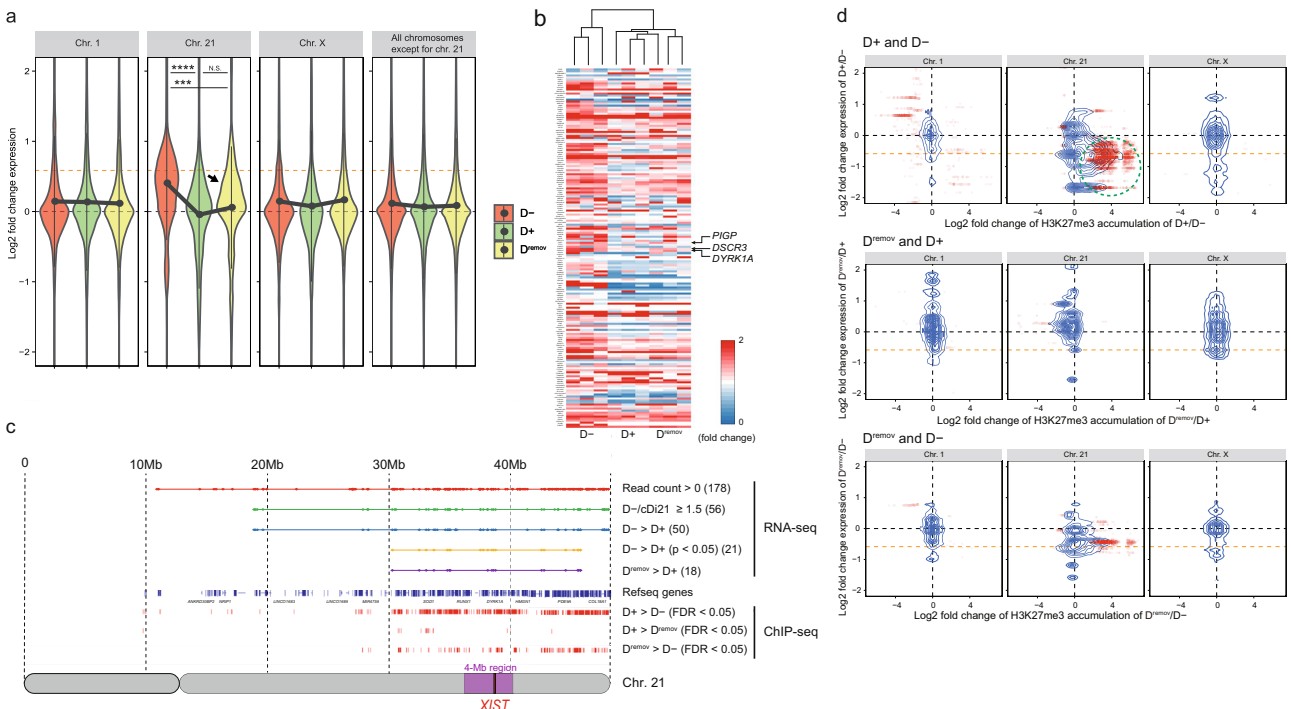

**Fig. 3 Forced *XIST* expression causes whole-chromosome inactivation, which is overall maintained after Dox removal. a** Violin plots showing relative log-transformed expression ratios of genes on each chromosome in *XIST*-Tri21 APC lines. Gene-expression levels (determined by RNA-seq) were normalised to those of cDi21 lines ($n = 3$ per cell line). The upper orange dashed lines indicate a ratio of 1.5, whereas the lower black dashed lines indicate a ratio of 1.0. Transcription from chromosome 21 was effectively suppressed in D+ cells, and inactivation was retained overall in D$^{remov}$ cells. A small hump was observed in chromosome 21 of a D$^{remov}$ cell line (black arrow). The plots show mean expression levels, with error bars indicating the SD. The data shown were analysed by the Kruskal–Wallis test with Bonferroni's correction. ***$P = 4.2 \times 10^{-5}$, ****$P = 5.1 \times 10^{-9}$; NS not significant ($P > 0.05$). **b** Heatmap with a hierarchical-clustering diagram depicting the transcriptional dynamics of chromosome 21. We analysed 178 genes with positive read counts on chromosome 21. Expression levels were normalised to those of cDi21 cell lines (Supplementary Data 1 and 2). Genes are ordered according to chromosomal position. **c** Overview of the RNA-seq and ChIP-seq data for chromosome 21. The distribution of 178 genes with positive read counts (red plots), 56 genes with an D−/cDi21-expression ratio ≥ 1.5 (light green), 50 genes in which Dox treatment decreased expression (blue), 21 genes in which Dox treatment significantly decreased expression (Welch's two-sample *t*-test, $P < 0.05$, yellow), and 18 genes in which Dox removal reverted the expression levels (purple). (Red bars) The distributions of regions with significantly higher accumulation of H3K27me3 are shown. False-discovery rates (FDRs) were calculated using the Benjamini–Hochberg method. $P$ and FDR < 0.05 were considered to reflect significant differences. **d** Integrated analysis of the ChIP-seq and RNA-seq data. H3K27me3 accumulation within each gene body and 2 kb upstream of the transcription start site were analysed. The density of all paired points is shown with a blue contour line. Genes with a significant difference in H3K27me3 accumulation between two lines are indicated (red points). The green dashed ellipse emphasises the distribution of inactivated genes with increased H3K27me3 accumulation after Dox treatment. The orange dashed line indicates a two-thirds decrease in gene expression.

(CRISPR)–CRISPR-associated protein 9 (Cas9). Short-tandem repeat (STR) analysis of targeted and non-targeted alleles in isolated iPSC colonies revealed that six types of genome-edited cell lines were obtained (single or double targeting of the M1, M2, or P alleles; Fig. 5b, c). *DYRK1A* expression in APCs decreased to similar levels found in DY$^{+/+/m}$-*XIST*-Tri21 APCs and were even lower in DY$^{+/m/m}$-*XIST*-Tri21 APCs than in Di21 APCs (Fig. 5d and Supplementary Fig. 11a). Accelerated APC proliferation proportionally decreased in both lines, and the double-targeted (DY$^{+/m/m}$) APC line showed more severe proliferative impairment, indicating that *DYRK1A* regulates APC proliferation in an expression level-dependent manner (Fig. 5e, f). Furthermore, both the DY$^{+/+/m}$ and DY$^{+/m/m}$ APC lines were subjected to *XIST*-mediated chromosome silencing. Given that *XIST* expression selectively inactivated the M2 chromosome in *cis*, these cell lines (with or without Dox treatment) exhibited various combinations of *DYRK1A* expression levels (zero, one, or two copies) and transcriptional karyotypes of chromosome 21 (trisomy or induced disomy) depending on the targeted alleles of the *DYRK1A* gene (ΔM1, ΔM2, ΔP, ΔM1ΔM2, ΔM1ΔP, or ΔM2ΔP; Fig. 5g). Dox treatment significantly reduced

*DYRK1A* expression by approximately half in ΔM1- and ΔP-DY$^{+/+/m}$ APCs, as expected, whereas no significant changes occurred in ΔM2-DY$^{+/+/m}$ APCs (decreased to 48.6%, 55.0%, or 99.1% of the levels in untreated cells, respectively; Supplementary Fig. 12a). Similarly, *DYRK1A* expression was not altered by chromosome silencing in ΔM1ΔM2- or ΔM2ΔP-DY$^{+/m/m}$ APCs (Supplementary Fig. 12b), suggesting that *DYRK1A* expression levels are precisely regulated in a gene dosage-dependent manner without being affected by the trisomy of other regions on chromosome 21 and that *XIST* exerts no *trans* effects on the M1 or P alleles of *DYRK1A*. Moreover, APC proliferation rates decreased by 22.8% and 22.5% after Dox treatment in the ΔM1- and ΔP-DY$^{+/+/m}$ lines, respectively (Fig. 5h, j). *XIST*-mediated silencing in the ΔM1ΔP-DY$^{+/m/m}$ APC line, in which all *DYRK1A* alleles were lost or suppressed, showed the most severe impairment of cell proliferation (27.4% reduction compared with untreated cells; Fig. 5i, k). To confirm the effect of *DYRK1A* on proliferation, APCs were treated with a DYRK1A inhibitor. Folding intermediate-selective inhibitor of DYRK1A (FINDY), which interferes with DYRK1A protein folding[32] (Supplementary Fig. 13a), significantly reduced

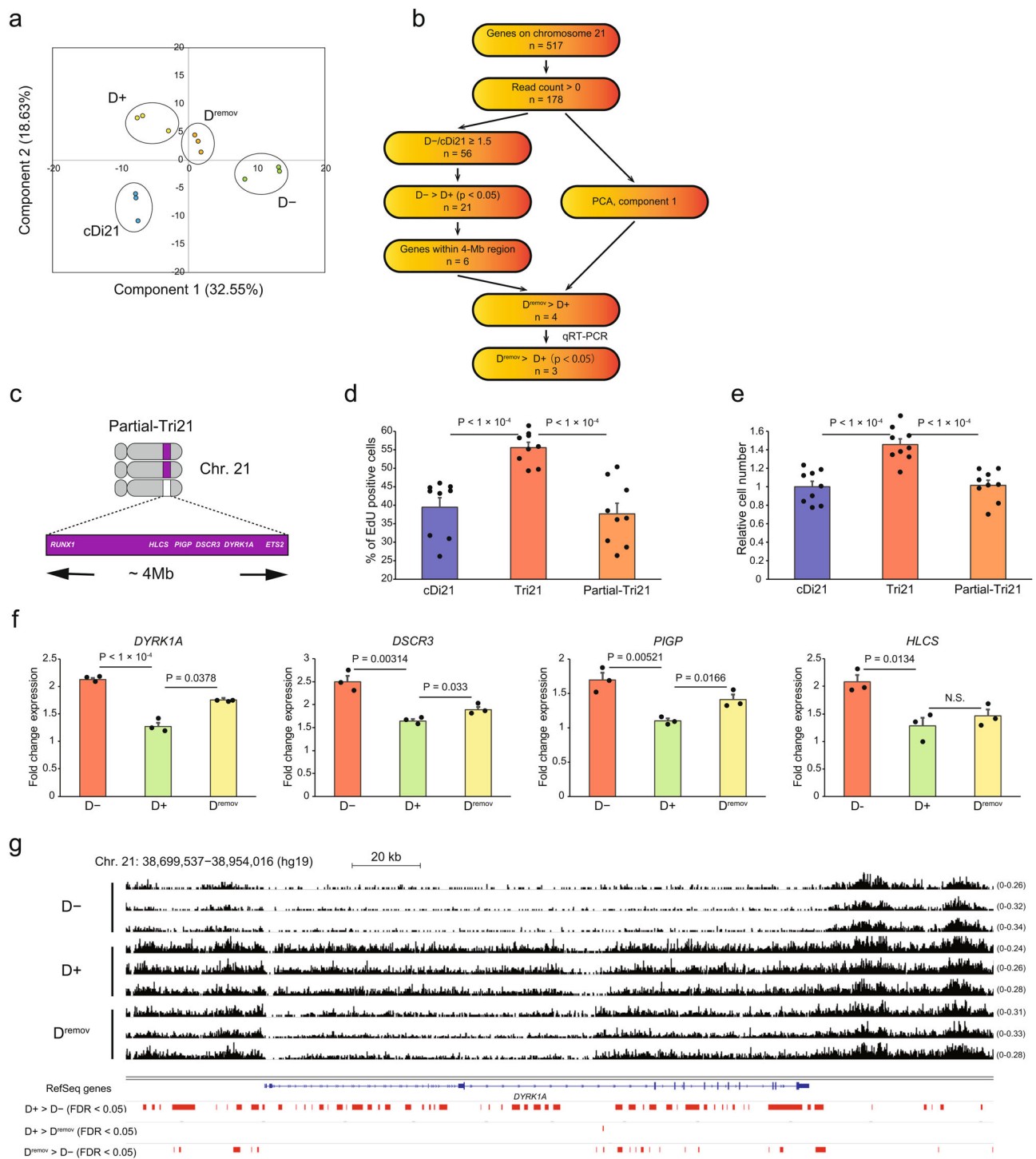

**Fig. 4 Identification of candidate marker genes for the aberrant proliferation of DS APCs using PCA and Partial-Tri21 cell line. a** Two-dimensional PCA of the expression profiles of 178 genes with positive read counts on chromosome 21 ($n = 3$ per cell line). Note that the values of the D$^{remov}$ cell lines in component 1 were relatively close to those of the D− cell lines. **b** Scheme for identifying genes responsible for APC overproliferation using RNA-seq data and PCA. **c** Schematic illustration of chromosome 21 in the Partial-Tri21 cell line. The 4-Mb region between *RUNX1* and *ETS2* was deleted from a single copy of chromosome 21. **d** The percentage of EdU-positive APCs ($n = 3$ independent differentiation experiments per cell line, each performed in triplicate). **e** Relative numbers of APCs 1 day after seeding. The cell numbers were normalised to that of the cDi21 cell lines ($n = 3$ independent differentiation experiments per cell line, each performed in triplicate). **f** Relative expression levels of candidate genes involved in the overproliferation of DS APCs ($n = 3$ experiments per cell line), normalised to that of cDi21 cell lines. **g** Map showing the distribution of H3K27me3 modifications in *DYRK1A* in *XIST*-Tri21 APCs. Integrative Genomics Viewer screenshot (http://software.broadinstitute.org/software/igv/) of H3K27me3 ChIP-seq track peaks on *DYRK1A* (hg19) in D−, D+, and D$^{remov}$ lines. The *Y*-axis shows the number of fragments per base pair per million reads. (Red bars) The distributions of regions with significantly higher H3K27me3 accumulation in the indicated cell lines. Data were obtained from three lines (**d**–**g**). The error bars represent the SEM. The data shown were analysed using Student's *t*-test (**d**–**f**). The FDR was calculated using the Benjamini–Hochberg method (**g**). NS not significant ($P > 0.05$).

**Table. 1 Component score and mean expression of genes located in a 4-Mb region on chromosome 21.**

| Gene | PCA component 1 Absolute value | D− | D+ | D$^{remov}$ |
|---|---|---|---|---|
| DYRK1A | 0.1195 | 1.69 | 1.03 | 1.30 |
| DSCR3 | 0.1193 | 1.70 | 1.12 | 1.27 |
| HLCS | 0.1188 | 1.62 | 0.88 | 1.11 |
| MORC3 | 0.1067 | 1.29 | 0.88 | 1.05 |
| PIGP | 0.1044 | 1.55 | 1.09 | 1.24 |
| CBR3 | 0.1004 | 2.24 | 1.52 | 1.51 |
| SETD4 | 0.0951 | 1.32 | 1.00 | 1.03 |
| TTC3 | 0.0910 | 1.33 | 0.64 | 0.96 |
| CHAF1B | 0.0859 | 1.76 | 1.03 | 1.03 |
| CBR3-AS1 | 0.0729 | 1.65 | 0.80 | 1.21 |
| LOC100506403 | 0.0687 | 1.04 | 1.00 | 1.00 |
| LINC00114 | 0.0687 | 1.04 | 1.00 | 1.00 |
| RUNX1 | 0.0662 | 1.41 | 0.63 | 0.98 |
| KCNJ15 | 0.0629 | 0.76 | 0.80 | 0.77 |
| CBR1 | 0.0521 | 0.55 | 0.47 | 0.43 |
| DOPEY2 | 0.0432 | 1.13 | 0.60 | 0.80 |
| SIM2 | 0.0389 | 2.96 | 1.60 | 2.27 |
| KCNJ6 | 0.0298 | 0.87 | 0.27 | 0.65 |
| DSCR9 | 0.0074 | 1.00 | 0.82 | 0.64 |
| LOC100133286 | 0.0053 | 1.00 | 1.07 | 1.10 |
| ERG | 0.0049 | 1.03 | 1.25 | 1.40 |
| ETS2 | 0.0045 | 0.90 | 0.69 | 0.63 |
| RIPPLY3 | 0.0025 | 1.03 | 0.93 | 1.09 |
| RUNX1-IT1 | 0.0012 | 1.00 | 1.04 | 1.24 |

The component score was obtained using principal component analysis (PCA) of genes within the 4-Mb region, while excluding genes with no read count in the RNA-seq data. Relative expression levels are shown and normalised to that of the cDi21 lines (n = 3 per cell line).

Tri21 APC proliferation without affecting that of Tri21 NPCs (Fig. 5l–o). These findings indicate that *DYRK1A* potently regulates APC proliferation in a dose-dependent manner.

There is evidence that ectopic overexpression of *Dyrk1A* induces the proliferation defect in NPCs through degradation of the cell cycle activator Cyclin D1 (refs. [33,34]). Tri21 NPCs showed impaired proliferation (Supplementary Fig. 14a, b) and low levels of Cyclin D1 (Supplementary Fig. 14c), as observed in *Dyrk1A* transgenic mice. Interestingly, Tri21 APCs showed a similar reduction in Cyclin D1 levels and increased expression of the CDK inhibitor p21$^{CIP1}$, in contrast to its proliferative phenotype (Supplementary Fig. 15). Furthermore, neither copy number reduction nor inhibitor treatment (Supplementary Figs. 11b and 13b) of *DYRK1A* altered these protein levels in APCs, indicating that *DYRK1A* regulates cell proliferation via different mechanisms in APCs and NPCs. The Janus kinase–signal transducer and activator of transcription (JAK–STAT) pathway is another main target of DYRK1A and is involved in gliogenic differentiation machinery in NPCs[14]. Consistent with the decreased expression of *DYRK1A*, Ser727-phosphorylated STAT3 was significantly decreased in not only DY$^{+/+/m}$ and DY$^{+/m/m}$ APCs but also FINDY-treated APCs and NPCs, whereas no changes in STAT3 protein levels were observed (Fig. 5p and Supplementary Figs. 13c, 16 and 17a, b).

Notably, the DY$^{+/+/m}$ APC line, which contains two copies of the normal *DYRK1A* allele in Tri21 line, retained higher proliferation rates than those of cDi21 (Fig. 5e, f), suggesting that correction of the *DYRK1A* gene dose is not sufficient to fully reverse APC overproliferation. On the other hand, APC proliferation was attenuated by Dox treatment in ΔM2-DY$^{+/+/m}$ APCs, despite the lack of altered *DYRK1A* expression (Fig. 5h, j). Similar results were obtained for M2-inactivation in ΔM1ΔM2- and ΔM2ΔP-DY$^{+/m/m}$ APCs, in which *DYRK1A* expression was unchanged after silencing

(10.1% and 11.9%, respectively; Fig. 5i, k). These results suggest the existence of other regulatory factors on chromosome 21 that control APC proliferation.

***PIGP*, not *DSCR3*, is involved in the proliferative pathophysiology of APCs in DS.** To investigate whether two other candidate genes (*PIGP* and *DSCR3*) are involved in the pathological features of DS, we transfected *piggyBac* transposon vectors containing human *PIGP* and/or *DSCR3* cDNA under regulation of the Tet-inducible system into NPCs differentiated from Partial-Tri21 iPSCs. Transduced NPC lines were then differentiated into the astrocyte lineage (Fig. 6a and Supplementary Fig. 18a). Expression levels of both *PIGP* and *DSCR3* were significantly increased in the corresponding Partial-Tri21 APC transfectants following Dox administration (Fig. 6b and Supplementary Figs. 18b and 19a). Proliferation rates of APCs, which were restored to normal levels after deletion of the 4-Mb critical region, increased by forced expression of *PIGP* + *DSCR3* or *PIGP* (Fig. 6c, d and Supplementary Fig. 18c, d) but not by single overexpression of *DSCR3* (Supplementary Fig. 19b, c). Because the dose-dependent activity of *PIGP* on cell proliferation was less effective compared with that of *DYRK1A*, we performed small-interfering RNA (siRNA)-mediated knockdown in Tri21 APCs to evaluate this effect. Knockdown of *PIGP*, but not *DSCR3*, slightly but significantly suppressed proliferation of Tri21 APCs (Fig. 6e–g and Supplementary Fig. 19d–f). Furthermore, *PIGP* overexpression in NPCs similarly enhanced proliferation (Supplementary Fig. 20a–c). Taken together, these results indicate that, together with *DYRK1A*, *PIGP* is another crucial molecule for APC proliferation that is involved in DS pathology.

## Discussion

Emerging evidence suggests that astrocytes play crucial roles in various pathological mechanisms in the central nervous system (CNS)[35,36]. Here, we focused on the aberrant proliferation of DS APCs, which is a major cause of astrocyte overpopulation in the brains of DS individuals, and explored essential regulators.

Selecting a suitable model system is crucial for exploring human diseases. A previous study using human iPSCs derived from several individuals with DS revealed transcriptional and epigenetic dysregulation in DS astrocytes[37]. Consistent with our results, *DYRK1A* expression was found significantly increased in both NPCs and astrocytes. Hierarchical clustering of the transcriptional dataset, however, failed to show an apparent segregation between DS and control lines, probably due to the inter-cell line variability. Indeed, human iPSCs from different donors are more divergent in terms of transcriptomes and cell phenotypes than those originating from different somatic cell types of the same donor, and significant variations in phenotypes among different iPSC lines hinders the precise analysis of human diseases[38]. On the other hand, several studies including ours have demonstrated that isogenic lines—derived from patients with constitutional mosaic for trisomy 21 (refs. [39,40]) or generated by genome-editing technologies[15,23,41]—mitigate the effects of genetic diversity, serving as critical controls for in vitro differentiation experiments. Therefore, we developed an isogenic cell model that combines *XIST*-mediated chromosome silencing and genome-editing technology using DS-specific iPSCs. All of these lines share a single genetic background, enabling a detailed comparative analysis of genotype–phenotype correlations between dosage-sensitive genes on chromosome 21 and proliferative dynamics in DS APCs.

Tet-regulated *XIST* is stably expressed after Dox treatment in both iPSCs and differentiated cells[20–22]. Nevertheless, *XIST* RNA was not detected in our differentiated *XIST*-Tri21 NPCs, likely due

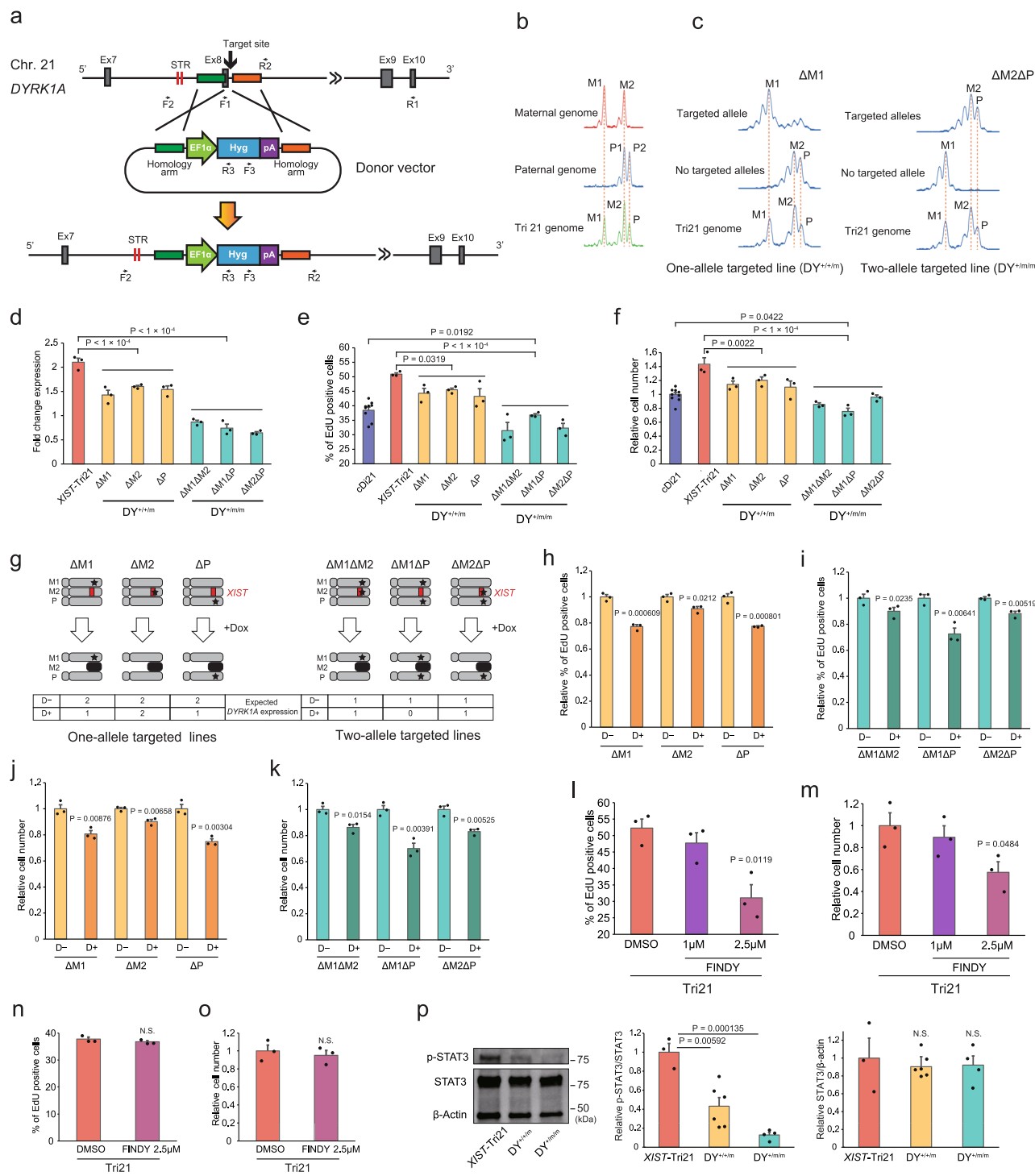

to silencing of the rtTA inserted in the *AAVS1* locus. Despite many descriptions of the *AAVS1* locus as a well-validated genomic safe harbour that enables stable expression of an inserted transgene[42], data from several studies have demonstrated that the *AAVS1* locus is not a genomic 'safe' as suggested, and that transgene expression varies in both iPSCs and differentiated cells due to DNA methylation[43,44]. Although the DNA-methylation status of the locus was not analysed in this study and unknown mechanisms may exist, additional transduction of the rtTA transgene using the *piggyBac* transposon system in NPCs led to efficient *XIST* RNA expression, followed by chromosome silencing.

We previously reported that chromosomes of trisomy cells exhibit a characteristic positional pattern in the nucleus, and two copies of maternal chromosomes 21 resulting from meiotic nondisjunction maintain their interaction[24]. Transcriptional analysis of a uniparental disomy line (M1 + M2 cDi21) revealed the existence of a small group of genes on chromosome 21 that were significantly upregulated only in the maternal alleles. To exclude the possibility of this parental-origin-specific effect on transcription, we selected two cDi21 lines (P + M1 and P + M2), in which one of the two maternal copies of chromosome 21 was selectively eliminated, as an isogenic control. In addition, *XIST*

**Fig. 5 Targeted deletion of *DYRK1A* promotes APC proliferation in a gene dosage-dependent manner. a** Schematic overview of the strategy used to target *DYRK1A*. **b** Comparative STR analysis of the *DYRK1A* locus was performed between a patient with DS and his parents' genomes. **c** STR analysis of *DYRK1A*-targeted iPSC lines to identify targeted alleles (F2/R3 primers) and non-targeted alleles (F2/R2 primers). **d** Relative expression levels of *DYRK1A* in DY$^{+/+/m}$- and DY$^{+/m/m}$-*XIST*-Tri21 APCs. The expression levels were normalised to that of cDi21 cell lines ($n = 3$ experiments per cell line). **e, f** Percentage of EdU-positive cells and the relative numbers of targeted APCs. Cell numbers were normalised to those of the cDi21 lines ($n = 3$ experiments for the cDi21 cell lines and the other cell lines, performed in triplicate). **g** Schematic illustration depicting the expected changes resulting from knocking out different *DYRK1A* alleles and chromosome silencing. Red rectangle, *XIST* transgene; black stars, *DYRK1A* target sites; black chromosomes, inactivated chromosome 21. The expected copy numbers of active *DYRK1A* genes in each APC line are shown. **h, i** Percentage of EdU-positive, *DYRK1A*-targeted cell lines, with or without Dox treatment. The percentage of EdU-positive cells was normalised to that of each Dox-untreated cell line ($n = 3$ experiments per cell line). **j, k** Relative cell numbers of the *DYRK1A*-targeted lines, with or without Dox treatment. The cell numbers were normalised to that of each Dox-untreated line ($n = 3$ experiments per cell line). **l–o** Percentage of EdU-positive cells and the relative numbers of Tri21 APCs (**l, m**) or Tri21 NPCs (**n, o**) after a 2-day treatment with FINDY ($n = 3$ experiments per cell line). The cell numbers were normalised to those of Tri21 APCs or NPCs without FINDY treatment ($n = 3$ experiments per cell line). **p** Immunoblot analysis of p-STAT3 (Ser 727) and STAT3 in *DYRK1A*-targeted *XIST*-Tri21 APCs. Expression levels were normalised to that of the *XIST*-Tri21 line ($n = 3–6$ experiments per cell line). Data were obtained from three lines (**p**). The error bars represent the SEM. The data shown were analysed by Student's *t*-test to compare two independent groups or one-way ANOVA with Bonferroni's correction (**d**, **e**, **f**). NS not significant ($P > 0.05$).

cDNA was inserted into one of the maternal copies of chromosome 21 (M2) in our study. Mean expression of chromosome 21 was restored to similar levels as in the cDi21 line by chromosome inactivation, and hierarchical-cluster analysis showed distinctive difference between D− and D+ lines, suggesting that chromosome silencing was successfully induced in our system. On the other hand, expression levels of *DYRK1A* in both Dox-treated *XIST*-Tri21 APCs and ΔM1-, ΔM2-, or ΔP-DY$^{+/+/m}$ APCs remained ~1.4-fold higher than those of cDi21 APCs (Figs. 4f and 5d). Given that we previously showed that *DYRK1A* expression in cDi21 iPSCs is the same as in healthy control iPSCs[24], this upregulation may have resulted from dosage effects of other genes, which were trisomic in DY$^{+/+/m}$ APC lines or escaped X-inactivation in *XIST*-Tri21 APCs.

To identify critical regulators of DS APC overproliferation, we focused on the variability of genetic responses after removal of *XIST*. In mouse embryonic stem (ES) cells, X chromosome inactivation (XCI) is reversible and depends on continued *Xist* expression; however, XCI is irreversible and independent of *Xist* after differentiation[27]. Chromosome inactivation can be introduced by ectopic expression of human *XIST* in somatic cells and maintained after removal of *XIST*, albeit to a different extent[20,45]. In agreement, the transcriptional levels of chromosome 21 were found stably suppressed for 3 weeks after Dox removal. Notably, a small subset of genes was rapidly restored from silenced disomic levels to reactivated trisomic levels, which was accompanied by a reoccurrence of the pathological phenotype in DS APCs. Such a multiplicity of gene reaction in XCI has been reported in several studies, indicating that 3–7% of mouse and 12–20% of human genes on the inactivated X chromosome escape XCI[46,47]. Likewise, ectopically expressed *XIST* RNA on autosomes can induce silencing in *cis*, whereas 15% of genes consistently escape from XCI and another 15% vary in terms of whether they are subject to, or escape from, inactivation[28,48]. Several bioinformatic studies have demonstrated that long-interspersed nuclear element repeats on the X chromosome or short-interspersed nuclear elements (such as *Alu* elements) are highly enriched around escape genes on autosomes, suggesting that genomic features contribute to the efficiency of *XIST*-mediated chromosome 21 inactivation[48,49].

Unexpectedly, our RNA-seq data showed *XIST*-mediated reduction of gene-expression levels in several other chromosomes, especially chromosome 18 (Supplementary Fig. 7). One possible explanation for this phenomenon is that trisomy of chromosome 21 may cause transcriptional alteration of a specific subset of genes on other chromosomes. It is known that the spatial organisation of chromosome territories in the cell nucleus is linked to genomic functions and regulation[50]. Gene-rich

chromosomes are located preferentially at the centre of the nucleus, whereas gene-poor chromosomes, such as chromosome 18, are located at its periphery[51]. Cell type-specific interaction patterns among chromosome territories are correlated with genome regulation at the global level, and the presence of a supernumerary chromosome 21 may perturb the physiological positioning of other chromosomes in the nucleus, leading to transcriptional dysregulation[24,52]. *XIST*-mediated silencing of extra chromosome 21 reversed trisomy-induced transcriptional changes in other chromosomes, leading to down-regulated expression levels, which was less noticeable but became evident after comparing the isogenic cell lines. Another possible explanation of the dysregulated transcription involves the high expression levels of *XIST* observed in our study. We introduced an additional rtTA into NPCs using the PB transposon vector, which generally provides robust transgene integration. After introducing the *XIST* transgene into mouse ES cells, silencing of autosomal genes occurred only in cell lines with high-copy transgenes, suggesting that dose-dependent regulation by *XIST* occurred[53]. Data from transgenic experiments indicated that *XIST* expression is essential, but not sufficient, for *XIST* RNA spreading and localisation[54,55]. Although *XIST* is a well-known *cis*-acting element, its *trans* effects on other chromosomes have not been reported. Nevertheless, its robust expression, which was higher than that observed physiologically, may explain the unexpected *trans* activity of *XIST*.

We identified two genes, *DYRK1A* and *PIGP*, as potent candidates responsible for the proliferative pathology of DS APCs. *DYRK1A* has been proposed to be closely involved in neural development, especially in fate specification and neuronal proliferation[56,57]. Studies conducted with human iPSCs and mouse models of DS exhibited impaired neural differentiation, which is improved by targeting *DYRK1A* pharmacologically or with short hairpin RNAs[58,59]. In addition, *Dyrk1a* overexpression promotes astrogliogenesis in mouse cortical progenitor cells by activating the STAT-signalling pathway[14], suggesting that DYRK1A plays a key role in cell-fate switching. Increased *DYRK1A* gene dosage attenuates neuronal proliferation rates[30], and loss of DYRK1A function triggers neural proliferation and cell death[56,57,60]. Moreover, DYRK1A extends the duration of the cell cycle G1 phase in a dose-dependent manner by reducing cyclin D1 and increasing p27$^{KIP1}$ (CDKN1B) expression[31,57]. Furthermore, DYRK1A can phosphorylate p53 and subsequently induce p53-target genes such as p21$^{CIP1}$, thereby impairing G1/G0–S-phase transition and attenuating proliferation[30]. These antiproliferative activities of DYRK1A are supported by the decreased number of neurons in humans and mouse models of

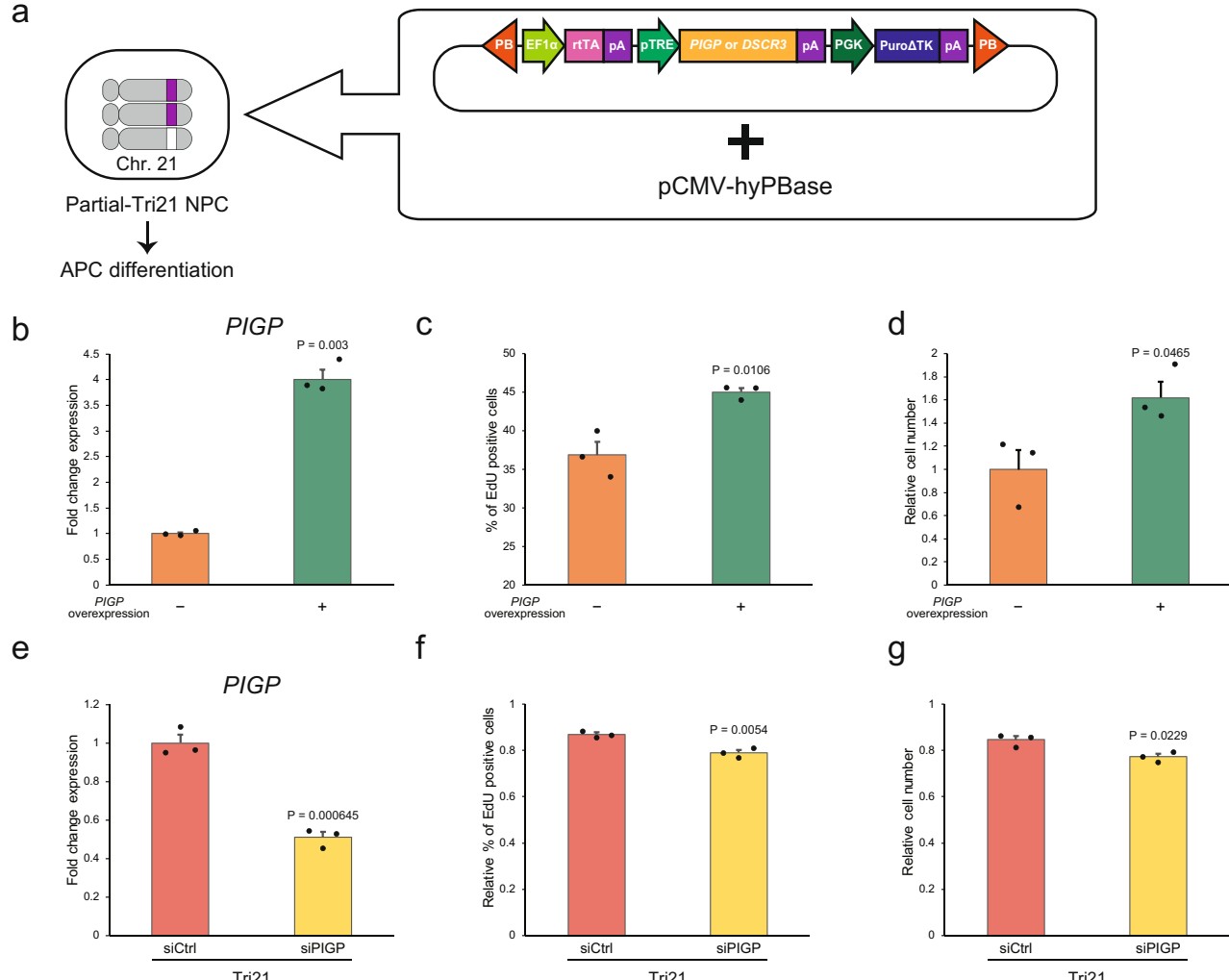

**Fig. 6 *PIGP* promotes DS APC proliferation in an expression level-dependent manner. a** Schematic depicting the transfection of a Dox-inducible *PIGP* or *DSCR3* transgene using a PB transposon vector and a hyperactive PB transposase into Partial-Tri21 NPCs. Note that chromosome 21 was trisomic except for the 4-Mb segment, which corresponds to the Down syndrome critical region. Transduced NPCs were differentiated to APCs. **b** Partial-Tri21 APCs stably expressing the *PIGP*-overexpression vector. *PIGP* overexpression was induced by a 6-week treatment with Dox. Expression levels were normalised to that of untreated APCs ($n = 3$ experiments per condition). **c, d** Percentage of EdU-positive cells (**c**) and relative cell numbers 1 day after seeding (**d**) in *PIGP*-overexpressing Partial-Tri21 APCs ($n = 3$ experiments per condition). Cell numbers were normalised to that of untreated APCs. **e** Relative *PIGP* expression in siRNA-treated Tri21 APCs. Expression levels were normalised to those in siCtrl-treated Tri21 APCs ($n = 3$ experiments per condition). **f, g** Percentage of EdU-positive cells (**f**) and relative cell numbers 1 day after seeding (**g**) in siRNA-treated Tri21 APCs ($n = 3$ experiments per condition). The percentage of EdU-positive cells and cell numbers were normalised to those of the untreated Tri21 line. siPIGP, PIGP siRNA. The error bars represent the SEM. The data shown were analysed by Student's $t$-test (**c–g**) or Welch's two-sample $t$-test (**b**).

DS[61,62]. In agreement, we observed impaired proliferation and attenuated Cyclin D1 expression in Tri21 NPCs. In contrast to its effects on neural precursors, we unexpectedly identified a proliferation-promoting activity of DYRK1A in APCs. Cyclin D1 levels were continuously down-regulated in Tri21 APCs and expression levels of Cyclin D1, p27[KIP1], and p21[CIP1] were not affected by DYRK1A deletion or inhibition, indicating that the dose-dependent actions of *DYRK1A* on cell cycle-associated proteins are cell context-specific.

STAT3 phosphorylation at Ser727 was altered by DYRK1A deletion or inhibition in APCs. In the CNS, STAT3 is highly expressed in astrocytes and is activated in response to multiple pathological stimuli such as ischaemia, spinal cord injury, or neurodegenerative diseases[63,64]. It remains unclear whether activated JAK–STAT signalling can stimulate astrocyte pro-liferation, especially during the physiological-developmental stage. Nevertheless, studies showed that proliferation of reactive

astrocytes in spinal cord injury is reduced by JAK inhibitors or conditional knockout of STAT3 (refs. [65,66]). Although the detailed mechanism remains to be elucidated, these indirect data and our current results suggest that increased expression of DYRK1A can stimulate APC proliferation via the JAK–STAT pathway.

*PIGP* is a component of the glycosylphosphatidylinositol–N-acetylglucosaminyltransferase (GPI–GnT) complex. More than 150 proteins have been identified as GPI-anchored proteins, which are expressed on the cell surface and anchored to the plasma membrane[67]. Biosynthesis of mammalian GPIs is initiated by the transfer of *N*-acetylglucosamine (GlcNAc) to generate GlcNAc-PI through the enzymatic activity of GPI–GnT. *PIGP* is one of seven GPI–GnT subunits, and mutations in *PIGP* that lead to reduced cell-surface expression of GPI-anchored proteins have recently been linked to early infantile encephalopathy[68]. Although the specific function of *PIGP* is unclear, its

overexpression has been reported to impair membrane localisation of Wnt-signalling receptors during embryogenesis[69]. Further studies are required to elucidate how PIGP accelerates APC proliferation in DS.

In this study, genes located outside the 4-Mb region, or whose expression levels were unchanged by Dox removal, were not verified in detail. Nevertheless, there are several candidate genes that may be involved in APC proliferation. $S100\beta$, a $Ca^{2+}$ sensor protein, is a critical regulator of DS pathophysiology that acts in a dose-dependent manner[35]. Previous reports demonstrated that $S100\beta$ interacts with the tumour suppressor p53 or activates the PI3K/Akt pathway, enhancing proliferation of various cell lines[70]. Notably, $S100\beta$ knockdown experiments demonstrated a significant decrease in DS astrocyte proliferation[11]. Further studies are required to elucidate the synergistic actions of $S100\beta$ with DYRK1A and PIGP in the accelerated proliferation of DS APCs.

Along with their high number, astrocytes in DS brains are morphologically more mature than in controls of corresponding age[9]. Nearly all differentiated cells in our protocol were positive for vimentin, and developmentally homogeneous and restricted to immature stage. No morphological differences were identified in our immature APCs between DS and controls, which was consistent with the previous observation. Further investigation using other differentiation protocols is warranted to identify the morphological changes in DS astrocytes.

Astrocytes support neuronal homoeostasis and regulate synaptic networks by promoting neuritogenesis and synaptogenesis[71]. However, DS astrocytes exert toxic effects on the formation and maturation of neural networks and neuron survival by reducing neuronal activity, inducing morphological alterations, and promoting neuronal apoptosis[11,19,72]. In the DS brain, astrocytes may act as important modulators in DS pathophysiology. Thus, identifying critical regulators for astrocyte overpopulation may be a critical first step for investigating disease mechanisms and developing new therapeutic strategies for DS. Our collection of isogenic iPSC lines will provide a useful resource for conducting detailed analyses of DS.

## Methods

**Human-iPSC generation and culture.** This study was approved by the Ethics Committee of Osaka University Graduate School of Medicine (approval number 13123-823). Informed consent was obtained from the patients' guardians in accordance with the Declaration of Helsinki. Human iPSCs were generated from a male patient with DS and cultured as previously reported[23]. Briefly, iPSCs were induced from the cord blood mononuclear cells of a male newborn with DS using a Sendai virus (SeV) vector encoding OCT4, SOX2, KLF4, and c-MYC[73]. iPSCs were maintained on mitomycin C (Merck)-inactivated mouse embryonic fibroblasts (MEFs) in human embryonic stem cell (hES) medium consisting of DMEM/F12 (Fujifilm Wako), KnockOut Serum Replacement (20%; Thermo Fisher Scientific), L-alanyl-L-glutamine (2 mM; Fujifilm Wako), MEM non-essential amino acid solution (1%; Fujifilm Wako), 2-mercaptoethanol (0.1 mM; Merck), and basic fibroblast growth factor (5 ng mL$^{-1}$; bFGF, Katayama Chemical) with or without Dox (2 μg mL$^{-1}$; Takara Bio). To remove SeV, siRNA L527 (Gene Design) was mixed with Lipofectamine RNAiMAX (Thermo Fisher Scientific) and used to transfect iPSCs as previously described[73]; complete removal of the SeV genome was confirmed using PCR and immunostaining with an anti-SeV-NP antibody. The iPSC cultures were passaged every 6–9 days, and the iPSCs used in this study were karyotyped by Chromocenter Inc. via G-band or Q-band analyses. Potency assays were conducted via immunocytochemistry. Briefly, iPSCs fixed in phosphate-buffer solution (PBS) containing 4% paraformaldehyde (Fujifilm Wako) were immunostained using primary antibodies against OCT4 (1:200; #sc-5279, Santa Cruz Biotechnology) and SSEA4 (1:200; #MAB4304, Merck), followed by the secondary Alexa Fluor 488- or Alexa Fluor 594-conjugated antibodies (1:500; Thermo Fisher Scientific). Three cDi21 lines, which were generated using a chromosome-elimination technique, were used as controls in this study[24]. Two cDi21 lines contained M1 maternal and paternal chromosomes (cDi21 M1 + P) and one line contained M2 maternal and paternal chromosomes (cDi21 M2 + P).

**Insertion of a Dox-inducible XIST transgene.** Full-length human XIST cDNA was kindly provided by Dr. Chikashi Obuse (Osaka University, Osaka, Japan). Dox-inducible XIST transgene was inserted into one copy of chromosome 21 in

Tri21 iPSCs using Cre recombinase-mediated cassette exchange (Supplementary Fig. 1a–c)[20]. A zinc-finger nuclease (ZFN) against the AAVS1 locus on chromosome 19 was designed to enable insertion of the 3G rtTA transgene, as described[74]. Insertion of a lox cassette into the DYRK1A locus was performed using the CRISPR–Cas9 system (Supplementary Table 4). Single-guide RNA (sgRNA) oligos for the CRISPR–Cas9 system were cloned into the BbsI sites of the pX330-U6-Chimeric_BB-CBh-hSpCas9 vector (#42230; Addgene). The pEF1α-3G rtTA-pA cassette was cloned from a pEF1α-Tet3G vector (#631167; Clontech). Full-length XIST cDNA with loxP and lox5171 sites was cloned into the pTRE3G vector (#631168; Clontech). On the day before transfection, iPSC colonies were dissociated into single cells using TrypLE Express (Thermo Fisher Scientific) in the presence of 10 μM ROCK inhibitor (Reagents Direct). Dissociated cells (1.0 × 10$^6$) were mixed with the donor vector (4–6 μg) and either a ZFN pair (left and right ZFNs, 0.5 μg each) or CRISPR–Cas9 (2 μg) and electroporated using the Neon Transfection System (settings: 1200 V, 20 ms, 2 pulses; Thermo Fisher Scientific). The electroporated cells were plated in 10-cm dishes with DR-4 IRR MEFs (Thermo Fisher Scientific). On day 4 post-electroporation, drug selection with G418 (150 μg mL$^{-1}$) or puromycin (0.5 μg mL$^{-1}$) was initiated according to the drug-resistance gene. The resulting colonies were selected on days 12–18. PCR-positive clones were further expanded. The lox cassette contained a positive/negative drug-selection marker (puroΔTK, encoding a fusion protein between the puromycin-resistance gene and a truncated version of herpes simplex virus type 1 thymidine kinase) between the loxP and lox5171 sites. For Cre recombinase-mediated cassette exchange, 1.0 × 10$^6$ cells were electroporated with a Cre expression vector (4 μg) and a donor vector (8 μg), which contained the XIST transgene with loxP and lox5171, as described above. The electroporated cells were plated, and then negative selection was initiated on day 4 post-transfection using 2 μM 1-[2-deoxy, 2-fluoro-8-d-arabinofuranosyl]-5-iodouracil. The resultant clones were analysed by PCR and positive clones were expanded.

**Maintenance and differentiation of NPCs.** NPC differentiation from iPSC lines (passage 40–50) was performed as previously described[15], with modifications. Briefly, embryoid bodies (EBs) were cultured for 8 days in Costar six-well, ultra-low-attachment plates (Corning) in hES medium without bFGF, consisting of 2 μM dorsomorphin (Merck), 10 μM SB431542 (Tocris Bioscience), and 10 μM ROCK inhibitor. Next, the EBs were attached for 13 days to Matrigel (Corning)-coated dishes in neuronal medium (N2B27 medium) consisting of DMEM/F12, Neurobasal Medium (Thermo Fisher Scientific), N2 supplement (1×; Thermo Fisher Scientific), B27 without vitamin A (1×; Thermo Fisher Scientific), GlutaMAX (1×; Thermo Fisher Scientific), L-alanyl-L-glutamine (2 mM; Fujifilm Wako), MEM non-essential amino acids solution (1%; Fujifilm Wako), 2-mercaptoethanol (0.1 mM), ROCK inhibitor (10 μM), and bFGF (20 ng mL$^{-1}$). Neural rosettes appearing in the centres of the attached EB colonies were isolated with TrypLE Express and replated in Matrigel-coated dishes in N2B27 medium containing bFGF (20 ng mL$^{-1}$). The culture medium (with or without 2 μg mL$^{-1}$ Dox) was changed every day and the cells were passaged every 3–5 days. To compensate for the insufficient expression of XIST RNA and accumulation of H3K27me3 in Dox-treated XIST-Tri21 iPSCs (Supplementary Fig. 3a–f) and NPCs—likely due to a lack of rtTA expression (Supplementary Fig. 4a, b)—rtTA was additionally transduced into XIST-Tri21 NPCs using the PB transposon vector and a hyperactive PB transposase (Supplementary Fig. 4a–f).

**Maintenance and differentiation of APCs.** We differentiated APCs from NPCs (passage 7–10) as previously described[75]. NPCs were dissociated with TrypLE Express, and 2 × 10$^4$ cells/well were plated on Matrigel-coated 24-well plates with Astrocyte Medium (ScienCell) supplemented with 10 μM ROCK inhibitor. The medium was changed every 2 days, and the cells were passaged every 4–6 days, with dissociation using TrypLE Express. In this study, APCs were passaged 7–8 times prior to analysis. When necessary, NPCs were differentiated into APCs by administrating Dox for 5 days. The Dox-treated cell lines were administered Dox for approximately 6 weeks, while the D$^{remov}$ cell lines were administered Dox for approximately 3 weeks, followed by culturing for an additional 3 weeks without Dox. Two XIST-Tri21 iPSC lines were generated from a male newborn with trisomy 21. Both iPSC lines were differentiated into NPC lines. The NPCs were independently transfected using the piggyBac vector encoding an rtTA to generate three lines. The NPC-derived APCs were subjected to qRT-PCR, cell-proliferation assays, RNA-seq, and ChIP-seq.

**Transfection of the piggyBac vector into NPCs.** To enhance rtTA, DSCR3, or PIGP expression, NPCs were transfected with a piggyBac vector harbouring an additional gene (rtTA, DSCR3, or PIGP) generated from the PB-TA-ERN vector (#80474; Addgene). The resulting vector (2 μg) and a pCMV-hyPBase vector (2 μg; a kind gift from the Sanger Institute) encoding transposase we co-transfected into NPCs (4.0 × 10$^6$) using the Neon Transfection System (settings: 1200 V, 20 ms, 2 pulses). After clone selection with puromycin (0.5 μg mL$^{-1}$), NPCs with the additional gene were established.

**Genome editing of DYRK1A using CRISPR–Cas9.** DYRK1A targeting was performed using CRISPR–Cas9. The sgRNA sequence was designed using CRISPR Direct (http://crispr.dbcls.jp/; Supplementary Table 4). The sgRNA oligos were

cloned into the pX330-U6-Chimeric_BB-CBh-hSpCas9 vector (#42230; Addgene). On the day before transfection, iPSC colonies were dissociated into single cells using TrypLE Express with 10 μM ROCK inhibitor. Cells were dissociated with TrypLE Express, after which $1.0 \times 10^6$ cells were mixed with CRISPR–Cas9 (2 μg) and the donor vector (6 μg), and electroporated using the Neon Transfection System (settings: 1200 V, 20 ms, 2 pulses). The electroporated cells were plated in 10-cm dishes with DR-4 IRR MEFs (Thermo Fisher Scientific). On day 4, drug selection with hygromycin (75 μg mL$^{-1}$) was initiated. The resulting colonies were selected on days 12–18. PCR-positive clones were further expanded. Primer sequences used for genome editing are listed in Supplementary Table 5.

**Transfecting siRNAs into APCs.** Cultured APCs were transfected with siRNAs using Lipofectamine RNAiMAX for 1 day. The final lipofectamine RNAiMAX and siRNA concentrations were 3 μL mL$^{-1}$ and 10 nM, respectively. siRNAs against *DSCR3* (#s20157; Thermo Fisher Scientific) and *PIGP* (#s27717; Thermo Fisher Scientific) were used. Silencer Select Negative Control siRNA #1 (#4390843; Thermo Fisher Scientific) was used as a control siRNA.

**Immunocytochemistry.** Immunocytochemistry was performed as previously described[41], with some modifications. Cells were fixed with 4% paraformaldehyde in PBS and permeabilised with 0.5% Triton X-100 in PBS for 15 min. After blocking with 5% foetal bovine serum in PBS for 30 min, the cells were incubated overnight at 4 °C with primary antibodies against H3K27me3 (1:200; #07-449, Merck), PAX6 (1:100; #130-095-598, Stemgent), SOX1 (1:100; #AF3369, R&D Systems), GFAP (1:1000; #Z033401, DakoCytomation), S100β (1:1000; #S2532, Merck), CD44 (1:200; #MABF580, Merck), or vimentin (1:500; #V2258, Merck). The cells were washed with PBS and incubated for 120 min with the appropriate Alexa Fluor 488- or Alexa Fluor 594-conjugated secondary antibodies (Thermo Fisher Scientific), followed by nuclei counterstaining with Hoechst dye (1:1000; #H342, Dojindo).

**RNA fluorescence in situ hybridisation (FISH).** RNA FISH was performed as previously described[24], with some modifications. *XIST*-Tri21 iPSC colonies were dissociated into single cells and cultured on coverslips for 2 days. The cells were fixed with 4% paraformaldehyde in PBS and incubated with ice-cold CSK buffer (100 mM NaCl, 300 mM sucrose, and 10 mM PIPES, pH 6.8) containing Triton X-100 (0.5%). After rinsing, the cells were dehydrated in ice-cold ethanol (100%) and air-dried. The *XIST* probe was labelled using a DIG-Nick translation Mix (Merck) according to the manufacturer's instructions. Hybridisation reactions consisting of labelled products (0.1 μg), herring sperm DNA (5 μg; Fujifilm Wako), baker's yeast transfer RNA (5 μg; Thermo Fisher Scientific), human Cot-1 DNA (3 μg; Thermo Fisher Scientific), and RNase Out (4 U μL$^{-1}$; Thermo Fisher Scientific) in hybridisation buffer [4× saline-sodium citrate (SSC) buffer, dextran sulfate (20%, w/v), bovine serum albumin (BSA); 4 mg mL$^{-1}$] were carried out overnight in a humidified incubator at 37 °C. The cells were stringently washed at 42 °C (three times in 2× SSC/50% formamide and three times in 2× SSC), and were blocked in 4× SSC containing 4 mg mL$^{-1}$ BSA and 0.1% Tween 20 at 37 °C prior to detection. Detection was performed in detection buffer (4× SSC containing 5% BSA and 0.2% Tween 20) with a mouse anti-digoxigenin antibody (#D8156, Thermo Fisher Scientific) for 50 min, followed by amplification with Cy3-conjugated antibody (#515-165-062, Jackson ImmunoResearch Labs) for 50 min. Nuclei were counterstained with 4′,6-diamidino-2-phenylindole (DAPI), and the slides were mounted in VECTASHIELD Antifade Mounting Medium (Vector Laboratories). Images were captured using an LSM 710 confocal scanning laser microscope (Carl Zeiss) and processed using Volocity software (PerkinElmer).

**qRT-PCR.** Total RNA was isolated from iPSCs, NPCs, and APCs with the NucleoSpin RNA II Kit (Macherey-Nagel). Reverse transcription was performed using ReverTra Ace qPCR RT Master Mix (Toyobo). qRT-PCR was performed using Thunderbird SYBR qPCR Mix (Toyobo). Gene-expression levels were normalised to β-actin (ACTB). All qRT-PCR primer sequences are listed in Supplementary Table 6.

**Single-cell cloning.** *XIST*-Tri21 iPSC colonies were dissociated into single cells using TrypLE Express, and then $1.0 \times 10^3$ cells were plated in a 10-cm dish with DR-4 IRR MEFs using iPSC culture medium containing 10 μM ROCK inhibitor, G418 (150 μg mL$^{-1}$) and Dox (2 μg mL$^{-1}$). On days 12–18, the resulting colonies were passaged to individual wells. These clones were expanded further, and on day 21 after Dox addition, they were fixed and assessed using immunocytochemistry.

**Cell-proliferation assay.** A Click-it EdU Alexa Fluor 488 Imaging Kit (Thermo Fisher Scientific) was used to measure cell proliferation. At 1 or 2 days before adding the thymidine analogue EdU, APCs were dissociated with TrypLE Express, and $5 \times 10^3$ cells/well were plated into a Matrigel-coated 96-well plate (Greiner Bio-one). On the day of EdU treatment, the cells were cultured with EdU (10 μM) for 8 h, followed by fixation with 4% paraformaldehyde in PBS and permeabilisation with PBS containing Triton X-100 (0.5%). The cells were stained as described by the manufacturer and then images were acquired using an In Cell Analyzer 6000 (GE Healthcare); EdU-positive cells were detected using the In Cell Developer

Toolbox 1.9 (GE Healthcare). Cell-counting assays were performed similarly as described above. Cells were fixed 1 or 2 days after cell plating ($5 \times 10^3$/well) in a Matrigel-coated 96-well plate and then nuclei were counterstained with Hoechst 33342 dye (1:1000). Images were acquired using the IN Cell Analyzer 6000, and stained nuclei were counted using the In Cell Developer Toolbox 1.9.

**DYRK1A inhibitor treatment.** The DYRK1A inhibitor FINDY (#SML1733; Merck) was dissolved in DMSO at 10 mM as a stock solution. Cells were then treated with FINDY (1 or 2.5 μM) or DMSO for 2 days, followed by cell-proliferation assays and western blot analysis. The final concentration of DMSO was 0.1% in all groups.

**RNA-seq.** APCs were harvested at passage 8 for RNA-seq, and total RNA was isolated from each sample using the NucleoSpin RNA II Kit. RNA-seq was performed by DNA Chip Research Inc. The integrity and quantity of total RNA were measured using an Agilent 2100 Bioanalyzer RNA 6000 Nano Kit (Agilent Technologies). Total RNA obtained from each sample was subjected to sequencing library construction using the NEBNext Ultra II Directional RNA Library Prep Kit for Illumina (New England Biolabs) with the NEBNext Poly(A) mRNA Magnetic Isolation Module, according to the manufacturer's instructions. Library quality was assessed using the Agilent 2200 TapeStation High Sensitivity D1000 ScreenTape System (Agilent Technologies). Pooled library samples were sequenced using a NextSeq 500 instrument (Illumina), with 76-bp single-end reads. Sequencing adaptors, low-quality reads, and bases were trimmed with the Trimmomatic 0.32 tool[76]. The sequence reads were aligned to the human reference genome (hg19) using TopHat 2.1.1 (Bowtie2 2.3.2)[77], which can adequately align reads (including splice sites) with the genome sequence. Files of the gene-model annotations and known transcripts were downloaded from the Illumina iGenomes website (http://support.illumina.com/sequencing/sequencing_software/igenome.html). These files were necessary for performing whole-transcriptome alignments with TopHat. The aligned reads were subjected to downstream analyses using StrandNGS 3.2 software (Agilent Technologies). The read counts for each gene and transcript (RefSeq Genes 2015.10.05) were quantified using the trimmed mean of *M*-value (TMM) method[78]. Plots were created using the ggplot2 (version 3.3.2) package in R 3.6 software.

**ChIP-seq.** For ChIP-seq, APCs were harvested at passage 8. Chromatin was prepared from the cells using the SimpleChIP Plus Enzymatic Chromatin IP Kit (Cell Signaling Technology) according to the manufacturer's instructions. Approximately $8 \times 10^6$ cells were used for each immunoprecipitation experiment. ChIP was performed using an antibody against H3K27me3 (#9733S; Cell Signaling Technology) and ChIP-seq analysis was performed by DNA Chip Research Inc. The quality and quantity of ChIP DNA and input DNA were measured with an Agilent 2100 Bioanalyzer High Sensitivity DNA Kit (Agilent Technologies). For each sample, 10 ng of DNA was subjected to sequencing library construction using the NEBNext Ultra II DNA Library Prep Kit for Illumina, according to the manufacturer's instructions. Library quality was assessed with the Agilent 2100 Bioanalyzer High Sensitivity DNA Kit. Pooled sample libraries were sequenced using NextSeq 500 in 76-bp single-end reads. Sequencing adaptors, low-quality reads, and bases were trimmed with the Trimmomatic 0.32 tool[76]. The sequence reads were aligned to the human reference genome (hg19) using Bowtie2 2.3.2 software[77]. PCR duplicates were removed using Picard ver. 1.119 (http://picard.sourceforge.net). The aligned reads were subjected to downstream analysis using the MEDIPS (version 1.30) package in R 3.4 software[79] (https://www.R-project.org/). We calculated the short-read coverage (extend value = 300) with genome-wide 100-bp bins using MEDIPS software. Differential coverage of the ChIP-seq data between groups of samples was detected based on the TMM method for genome-wide 100-bp sliding windows.

**STR analysis.** To perform STR genotyping, DNA was extracted from iPSCs using a DNeasy Blood & Tissue Kit (Qiagen). To assess *DYRK1A*-targeted alleles, junctional PCR (homologous recombination+) and outside PCR (homologous recombination−) were performed using KOD FX Neo enzyme solution (Toyobo). PCR products or genomic DNA were used for PCR with PrimeSTAR MAX (Takara Bio), a fluorescently labelled forward primer, and a reverse primer. The final PCR products were mixed with an internal lane standard 600 (Promega) and HiDi formamide (Thermo Fisher Scientific) and separated by capillary electrophoresis on an ABI 310 Genetic Analyzer (Thermo Fisher Scientific), according to the manufacturers' instructions. Primer sequences are listed in Supplementary Table 5.

**Allele-specific SNP-silencing analysis.** Total RNA was isolated from *XIST*-Tri21 APCs (Dox-untreated and treated cell lines) using the NucleoSpin RNA II Kit. Reverse transcription was performed using ReverTra Ace qPCR RT Mix. With the resulting total cDNA, PCR was performed using primers that amplified a region containing an SNP (rs457705) on exon 8 of *ETS2* on chromosome 21. *ETS2* primer sequences used are provided in Supplementary Table 7.

**Western blotting.** Cells were lysed with RIPA Buffer (Fujifilm Wako) containing a mixture of protease (Merck) and phosphatase (Nacalai Tesque) inhibitors. Lysates containing equal amounts of protein were mixed with an appropriate amount of

Laemmli sample buffer (2×; Bio-Rad Laboratories) and 2-mercaptoethanol and denatured at 95 °C for 5 min. The samples were separated using 7.5 or 15% sodium dodecyl sulfate-polyacrylamide gel electrophoresis and transferred to poly-vinylidene fluoride membranes (Bio-Rad Laboratories). The membranes were washed with Tris-buffered saline containing Triton X-100 (0.05%) and incubated with Blocking One or Blocking One-P (Nacalai Tesque) buffer for 60 min. Mouse anti-STAT3 (1:1000; #9139; Cell Signaling Technology), rabbit anti-Phospho-STAT3 (Ser727; 1:500; #9134; Cell Signaling Technology), rabbit anti-DYRK1A (1:1000; #2771; Cell Signaling Technology), rabbit anti-Cyclin D1 (1:1000; #55506; Cell Signaling Technology), rabbit anti-p27$^{KIP1}$ (1:1000; #3686; Cell Signaling Technology), and rabbit anti-p21$^{CIP1}$ (1:1000; #2947; Cell Signaling Technology) were used as primary antibodies. Horseradish peroxidase-conjugated anti-rabbit or anti-mouse IgG antibodies (1:2500; #W401B, #W402B; Promega) were used as secondary antibodies. As control, β-actin was detected with anti-β-actin pAb-HRP-DirecT (1:2000; #PM053-7; MBL). Blots were developed using Clarity Western ECL Substrate (Bio-Rad Laboratories) and visualised using ImageQuant LAS 4000 Biomolecular Fluorescence Image Analyzer (GE Healthcare). When necessary, the antibody was stripped with Restore Western Blot Stripping Buffer (Thermo Fisher Scientific). Quantification was performed using ImageJ software (http://imagej.nih.gov/ij/). Full immunoblot images are shown in Supplementary Fig. 21.

**Statistics and reproducibility**. All statistical analyses were performed using EZR software. Comparisons of two groups were performed using Student's t-test or Welch's two-sample t-test. We evaluated multiple comparisons using one-way analysis of variance (ANOVA) or the Kruskal–Wallis test with Bonferroni's correction. P values <0.05 were considered statistically significant. The data are presented as the mean ± standard error of the mean (SEM) or standard deviation (SD).

**Reporting summary**. Further information on research design is available in the Nature Research Reporting Summary linked to this article.

## Data availability

The RNA-seq and ChIP-seq data reported here have been deposited in the DDBJ Sequenced Archive under accession numbers DRA010528 and DRA010529, respectively. Source data underlying plots shown in figures are provided in Supplementary Data 3. The generated iPSC lines will be made available to the scientific community. All other data are available from the corresponding author upon reasonable request.

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

## Acknowledgements

We would like to thank Professor Chikashi Obuse for generously providing us with human *XIST* cDNA. We are grateful to Professor Judy Noguchi for carefully reading the manuscript and providing critical comments. We also thank the staff at the Center for Medical Research and Education, Graduate School of Medicine, Osaka University for providing technical support. This work was supported by the Practical Research Project for Rare/Intractable Diseases from the Japan Agency for Medical Research and Development (AMED) (grant number JP19bm0804009 to Y.K.) and JSPS KAKENHI (grant numbers JP16K10090 and JP19H03619 to Y.K.; JP17K16257 to K.K.).

## Author contributions

K.K. and Y.K. conceptualised and designed the experiments and wrote the manuscript. K.K. conducted most of the experiments and contributed to the data analysis. T.N., N.N., H.K., H.Y., K.B. and K.H. conducted differentiation experiments. A.T. and K.S. assisted with RNA-seq and ChIP-seq analysis. H.T., H.A., and Y.K. collected clinical data to characterise patients for iPSC establishment. K.O. and Y.K. supervised the entire project. All authors reviewed and commented on the manuscript.

## Competing interests

The authors declare no competing interests.

## Additional information

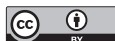

