## [Peer Review File · Communications Biology]

Reviewers' comments:

Reviewer #1 (Remarks to the Author):

The manuscript of Kawatani et al. reports genotype phenotype correlations in a panel of Down syndrome isogenic astrocytes derived from iPSCs. They used the XIST-mediated transcriptional silencing system inserted in one copy of chromosome 21 published in 2013 where the expression of XIST is regulated by tetracycline. They used additional iPSCs carrying deletions of the Down syndrome critical region that they published earlier and developed additional iPSCs with two copies of chromosome 21 candidate genes. Using cell proliferation assays with EdU incorporation, the authors found that astrocyte precursors carrying an extra copy of chromosome 21 show higher proliferation rate and that this effect was reversed by doxycycline-induced XIST expression and chromosome 21 silencing.

They further show the implication of the Down syndrome critical region and narrow down to three single genes. Firstly, they demonstrate the importance of DYRK1A silencing on reversion of astrocytes over proliferation. Second, they show similar effects using a DYRK1A inhibitor effective at the highest dose (2.5 μ M). Finally, they show that PIGP but not DSCR3 are involved in over proliferation of the astrocytes and confirm using siRNA against PIGP.

The results presented are convincing and the use of additional methods (siRNA or specific chemical inhibitors) to confirm the role of specific genes strengthen the findings.

Additional data on astrocyte morphology would be interesting to present since changes have been reported in Down syndrome brains (see missing references below). In addition, improvement of writing could facilitate the reading of the manuscript, going straight to the points. Results section could be shortened (some technical details could be moved in the Mat & Meth section) and parts of the discussion are missing.

Major points:

1. The authors modified the XIST protocol initially published in order to compensate for the lack of expression of XIST in neural precursor cells in their hands, and expressed the reverse tetracyclin trans activator using a piggyback. This part of the paper could be shortened or moved in the Mat & Meth. It would be important to provide some explanation and comment on differences in protocols.
2. The 4Mb deletion model was previously published in Cell Reports in 2016 by the authors (Banno et al.). This part does not need to be detailed then.
3. The rationale on single genes selection could be better explained with details on what should be seen in Supplementary Table 3.
4. It would be important to discuss S100 β since it is highly expressed in astrocytes, overexpressed in Down syndrome and has been shown to be a key gene in the astrocyte phenotype (see the review of Cresto et al. Trends Neurosci. 2019 PMID 31300246).
5. There are important references missing:
Zdaniuk et al. Folia Neuropath. 2011 PMID 21845539 showing a substantial increased number of glial fibrillary acid protein positive cells in all age range samples of Down syndrome brains and a change in the morphology of astrocytes.
Jorgensen et al. J. Neurol. Sci. 1990 PMID1977892 showing astrocytic reaction
Ponroy Bally et al. HMG 2020 PMID 31943018 this recent paper used iPSCs from 3 controls and 4 individuals with Down syndrome which are not isogenic but provide different genetic backgrounds while the present study relies on one genetic background only. A comparison of their results with the present results is mandatory and will be interesting. The use of mosaic isogenic iPSCs needs also to be discussed (several papers).

Minor changes:

p. 7: There is a problem with the phrasing of the second paragraph. Lines 10-15 are not necessary here.

p. 9 first paragraph: how long is Dox treatment?

p. 9: second paragraph: why not using SOX9? Also S100 β is a protein encoded by a gene from chromosome 21.

p. 12 second paragraph: problem with phrasing.

Mat Meth: there is no details on DYRK1A inhibitors.

Reviewer #2 (Remarks to the Author):

In the manuscript, the group of Kitabatake reported the making and the use of a new panel of DS isogenic iPSC cell lines. They used the XIST-mediated inactivation to generate control pseudo-disomic lines and Crisp/cas9 tools to manipulate candidate genes of interest. Using this panel, the authors analysed the consequence of DS on astrocyte precursor proliferation and differentiation. They reported that trisomic iPSC cells displayed increased proliferation of Astrocyte Precursor cells (APC) and that induction of XIST in XIST-Tri21 iPSC-derived APC can rescue the phenotype. The XIST-mediated chromosome inactivation was maintained over 3 weeks. The authors also demonstrated that the trisomy of the Runx1-Ets2 region is necessary for having an increase in the APC proliferation rate and that DYRK1A dosage is necessary for this effect with a slight effect of PIGP. The results are of interest, but they may need further clarification before publication.

Major comments:

1. The authors refer to the "generated a patient-derived Tri21 iPSC line that contains one paternal copy and two maternal copies of chromosome 2122". According to the Sup fig 2 the two maternal copies are different. Thus if they are different what is the chromosome left in the "corrected disomy 21 iPSC line (cDi21 iPSC)"?
2. Furthermore, it would be of interest to know if the insertion of the XIST complementary DNA (cDNA) happened the XIST-Tri21 iPSC either in one of the maternal or the paternal copies of chromosome 21. This is indeed indicated in sup fig10 and discussed lane 243 but should be inserted in the first paragraph of the results while presenting the XIST-TRI21 iPSC cells. At this point, the authors should discussed how their "corrected disomy 21 iPSC line (cDi21 iPSC)" is comparable to the activated XIST-Tri21 iPSC.
3. A clear description of all the panel of DS isogenic iPSC cell lines should be made available in a table. The number of passages from the initial Trisomic cells should be mentioned for all subclones.
4. In addition, it should be clearly indicated where all the IPS cells are made available to the scientific community.
5. In the figure 2 panel F, the authors reported the fold change expression of CCT8, RCAN1 , DYRK1A and C21orf33 which is above the expected 1.5x fold increase usually observed in DS cells compared to disomic controls. Can the authors provide some explanation?
6. The panel B in figure 3 is supposed to show "178 genes with positive read counts on chromosome 21" (lane 181). Unfortunately, it is not possible to read the gene name on the figure. Thus first, the figure 3B should be made available in a larger format and a table should be included in the supplementary materials with all the FC details (Finally I found it in the Sup table 1 but no reference in the text between lane 174-190). At present although I can partially share the conclusion that some genes recovered a normal expression pattern in D+ or DRemov condition, but certainly not all of them... additional question should be raised here.
7. Why the fold change of DYRK1A is shown on fig 3f and 4f? are the data for D- and D+ the same ? If yes remove from 3F.
8. Lane 235: the authors stated "DYRK1A was identified as the most potent candidate gene in the PCA list (Supplementary Table 2)". In this table DYRK1A is ranked 7... thus Can the authors explain why this is the most potent candidate in the PCS list and which criteria was/were used?
9. Figure 5e it would be better to represent P-STAT3/STAT3 and then STAT3/b-actin.
10. Can the authors explained or proposed a mechanism for the observation reported lane 181: "Similarly, DYRK1A expression was not altered by chromosome silencing in $\Delta M1\Delta M2$ - or $\Delta M2\Delta P$ -DY+/m/m APCs (Supplementary Fig. 12b)."?
11. In the experiment to demonstrate the role of DYRK1A, the expression level of the gene in XIST-TG21 iPSC showed a 2x increase. When DYRK1A is targeted, DeltaM1, DeltaM2 or Delta P, the level was still around a 1.5x increase. Can the author propose some explanation? Are the control disomic cells a good control? May be it can explain the observation reported lane 266-269.
12. Do the authors have a possibility to detect DYRK1A protein level and to measure DYRK1A kinase activity in their mutant cells and when they treated with FINDY? This will be more direct compare to looking at expression level.
13. Overall, all the results should be presented with individual data points rather than histogram to allow a clear view of the variability and of the number of points used in each experiments.

Miscellaneous

1. Lane 124: continuously to continuously
2. The sex of the IPS should be clearly defined (male or female)

Reviewer #3 (Remarks to the Author):

In this manuscript Kawatani et al. study the genetic basis of a characteristic brain feature in Down syndrome, which is the overabundance of astrocytes. To this end, the authors perform proliferation assays on astrocyte precursor cells (APCs) differentiated from Tri21-iPSC-derived NPCs and gene expression analysis (RNA-seq and ChIP-seq) in these cells. To reduce noise from genetic variation they use a panel of engineered isogenic iPSC lines with different number of Chr.21 genes (disomy 21, trisomy 21, and partial trisomy 21) and trisomy 21 cell lines with XIST insertion for chromosome silencing. They compare the transcriptional profiles of XIST-mediated silencing of Tri21-APC lines in 2 experimental settings and identify a set of Chr.21 genes potentially involved in APC proliferation. By comparing the transcriptome profiles of additional cell lines, they narrowed down this list of genes. Finally, they assay the dosage-dependent role of two of these genes, DYRK1A and PIGP, on APC proliferation.

The manuscript is beautifully written and the summary of the experimental setting is clearly depicted in the first figure, which facilitates the reading of the results. With few exceptions, experiments are well controlled.

The main interest of the work is twofold; it provides new cellular models to investigate Down syndrome and identifies two Chr.21 genes involved in a relevant phenotype for the intellectual disability and probably other neurological features associated to Down syndrome. DYRK1A overexpression has already been implicated in the astroglial phenotype associated to Down syndrome (Kurabayashi et al. EMBO reports 2015). Nonetheless, the identification of DYRK1A as an important gene for APC proliferation is of interest in the fields of cell biology and developmental biology. The possible role of PIGP in cell proliferation is novel but needs further investigation.

Comments:

1. The experiments indicate that the overexpression of DYRK1A increases APC proliferation. However, evidences obtained in animal models and cell cultures have shown that DYRK1A overexpression reduces the proliferation of NPCs. Did the authors perform proliferation assays in the different NPGs lines before culture them in the "astrocyte medium"? As mention in the Discussion, the attenuated proliferation of progenitors overexpressing DYRK1A has been attributed to alterations in Cyclin D1, p27 and p21 levels. It could be possible that the effect of DYRK1A on the stability of these cell cycle regulators is cell context-specific. Expression data of these proteins in both Tri21-NPCs and their derivative APCs and EdU incorporation data in Tri21-NPCs treated with the DYRK1A inhibitor FINDY should be easy to generate and could clarify this apparent contradiction.

The results in the EMBO Reports paper from Kurabayashi et al. show that the overexpression of DYRK1A increases the astrogligenic potential of brain NPCs during development. The authors could add a paragraph in the Discussion explaining the implications of their results in astrogligenesis.

2. In Fig. 2b, the authors show immunostainings for 4 astroglial markers in XIST-Tri21 APCs cultures. It is possible that the percentage of cells expressing each of these markers varies depending on the differentiation stage of the culture, which could be influenced by the overexpression of DYRK1A and/or other Chr.21 genes. Did the authors check if the percentages of cells expressing these markers are maintained among the conditions analysed (D-, D+ and Dremov)? This information could be relevant for the interpretation of the results and should be included in the manuscript.

3. PIGP encodes a protein of the glycosylphosphatidylinositol-anchor biosynthesis pathway with no reported functions on cell proliferation. The proliferation defects in the gain- and loss-of function experiments presented in Fig. 6 are moderate. The authors should perform additional experiments to support the dosage-dependent effect of PIGP on APC proliferation (i.e. same type of

experiments in an additional cell line, cDI21 iPSCs for example).

Methodology:

- The number of passages of the different cell lines used in each experimental procedure should be indicated in Methods.
- The numbers of generated cell lines are in Fig. 1 (legend). The number of independent cell lines used in each experiment should also be clearly state in Methods and figure legends.
- The same for the statistical test used to compare different sets of data.

Minor comments:

The sentence "In the DS brain, astrocytes may act as a primary effector in DS pathophysiology" (page 26) is an overstatement.

Point-by-point responses to the Reviewers' comments

We are grateful to the editor and reviewers for their constructive comments and insightful suggestions that have helped us to improve our manuscript. As indicated in the responses below, we have taken all comments into consideration for the revision of our manuscript. For clarity, changes are highlighted in red in the revised manuscript.

=====

Reviewer #1.

The results presented are convincing and the use of additional methods (siRNA or specific chemical inhibitors) to confirm the role of specific genes strengthen the findings. Additional data on astrocyte morphology would be interesting to present since changes have been reported in Down syndrome brains (see missing references below). In addition, improvement of writing could facilitate the reading of the manuscript, going straight to the points. Results section could be shortened (some technical details could be moved in the Mat & Meth section) and parts of the discussion are missing.

Response: Thank you for your positive assessment. We hope that the revised manuscript is now acceptable for publication in *Communications Biology*.

Major points:

1. The authors modified the XIST protocol initially published in order to compensate for the lack of expression of XIST in neural precursor cells in their hands, and expressed the reverse tetracyclin trans activator using a piggyback. This part of the paper could be shortened or moved in the Mat & Meth. It would be important to provide some explanation and comment on differences in protocols.

Response: Thank you for this suggestion. We have shortened the rtTA transduction part in the Results (**page 8, lines 114–121**) and clarified it in the Methods.

2. The 4Mb deletion model was previously published in Cell Reports in 2016 by the authors (Banno et al.). This part does not need to be detailed then.

Response: As suggested, we have shortened the description of the 4Mb deletion model (page 13, line 208).

3. The rationale on single genes selection could be better explained with details on what should be seen in Supplementary Table 3.

Response: We apologize for our unclear description. First, 178 genes with positive expression on chromosome 21 were listed, after which the 24 genes located in the 4-Mb region were ranked in order of the PCA component 1 values. Because deletion of this 4-Mb region from chromosome 21 led to a significant decrease in the proliferation rate of APCs, we speculated that responsible genes for accelerated APC proliferation in DS are located within this 4 Mb region. *DYRK1A* was ranked in the top position among the 24 genes (also in the 7th position among the 178 genes), and was thought to be the most potent candidate.

We modified Supplementary Table 3 and moved it to the main text as Table 1. Moreover, the gene selection method was described in detail (page 13–14, lines 211–220).

4. It would be important to discuss *S100β* since it is highly expressed in astrocytes, overexpressed in Down syndrome and has been shown to be a key gene in the astrocyte phenotype (see the review of Cresto et al. Trends Neurosci. 2019 PMID 31300246).

Response: Thank you for this comment, and we are in agreement. As *S100β* is a critical factors located on chromosome 21, we have discussed it in the Discussion (page 27, lines 458–467).

5. There are important references missing:
Zdaniuk et al. Folia Neuropath. 2011 PMID 21845539 showing a substantial increased number of glial fibrillary acid protein positive cells in all age range samples of Down syndrome brains and a change in the morphology of astrocytes.
Jorgensen et al. J. Neurol. Sci. 1990 PMID1977892 showing astrocytic reaction
Ponroy Bally et al. HMG 2020 PMID 31943018 this recent paper used iPSCs from 3

controls and 4 individuals with Down syndrome which are not isogenic but provide different genetic backgrounds while the present study relies on one genetic background only. A comparison of their results with the present results is mandatory and will be interesting. The use of mosaic isogenic iPSCs needs also to be discussed (several papers).

Response: We have cited these three papers (references 9, 37, and 38) in the Discussion (page 19–20, lines 323–340).

Minor changes:

1. p. 7: There is a problem with the phrasing of the second paragraph. Lines 10-15 are not necessary here.

Response: We shortened this section and moved it from the Results to the Methods (page 7, lines 108–110; page 28, lines 482–483).

2. p. 9 first paragraph: how long is Dox treatment?

Response: Doxycycline was administered for 5 days. We have clarified this in the Results (page 8, lines 120–121).

3. p. 9: second paragraph: why not using SOX9? Also S100 β is a protein encoded by a gene from chromosome 21.

Response: We used GFAP and S100 β as astrocyte markers (page 9, lines 133–134). We did not use SOX9 as an astrocyte marker as some studies reported that SOX9 is also expressed in neural stem cells in addition to astrocytes (Scott et al., 2010, *Nat Neurosci*).

4. p. 12 second paragraph: problem with phrasing.

Response: We apologize for the ambiguous wording, which has been accordingly revised

(page 12, lines 186–190).

5. Mat Meth: there is no details on DYRK1A inhibitors.

Response: We apologize for this oversight. We have accordingly described the DYRK1A inhibitors in the **Supplementary Methods**.

=====
Reviewer #2.

Major comments:

1. The authors refer to the “generated a patient-derived Tri21 iPSC line that contains one paternal copy and two maternal copies of chromosome 2122”. According to the Sup fig 2 the two maternal copies are different. Thus if they are different what is the chromosome left in the “corrected disomy 21 iPSC line (cDi21 iPSC)”?

Response: Thank you for this insightful comment. As shown in Supplementary Figure 2c, our Tri21 iPSC line contains one paternal (P) and two maternal (M1 and M2) copies of chromosome 21, which are derived from meiotic nondisjunction. Because most cases of trisomy 21 are caused by nondisjunction in the first meiotic division, these two copies of maternal chromosome 21 were inherited from the maternal grandfather and grandmother. In our previous study, we generated three types of cDi21 lines (M1+P, M2+P, and M1+M2; Omori et al. 2017 *Sci Rep*) and used two cDi21 lines that contain M1 maternal and paternal chromosomes (cDi21 M1+P) as well as another line that contains M2 maternal and paternal chromosomes (cDi21 M2+P) in this study. We have clarified this in the Methods (page 28, lines 482–486).

2. Furthermore, it would be of interest to know if the insertion of the XIST complementary DNA (cDNA) happened the XIST-Tri21 iPSC either in one of the

maternal or the paternal copies of chromosome 21. This is indeed indicated in sup fig10 and discussed lane 243 but should be inserted in the first paragraph of the results while presenting the XIST-TRI21 iPS cells. At this point, the authors should discuss how their “corrected disomy 21 iPSC line (cDi21 iPSC)” is comparable to the activated XIST-Tri21 iPSC.

Response: We agree that this is an important point to discuss. We previously reported that chromosomes of trisomy cells exhibit a characteristic positional pattern in the nucleus, and that two copies of maternal chromosomes 21 resulting from meiotic nondisjunction conserved their interaction (Omori et al. 2017 *Sci Rep*). In the uniparental disomy line (M1+M2 cDi21), some genes on chromosome 21 showed significantly upregulated expression. To exclude the possibility of this parental-origin-specific effect on transcription, we used P+M1 and P+M2 cDi21 lines as an isogenic control in this study. In addition, XIST cDNA was inserted into one of the maternal copies of chromosome 21 (M2). Mean expression of chromosome 21 was restored to similar levels as those in the cDi21 line by chromosome inactivation, and hierarchical-cluster analysis showed a distinctive difference between D⁻ and D⁺ lines, suggesting that chromosome silencing was successfully induced in our system.

We moved the description regarding SNP analysis for identifying XIST-inserted chromosomes to the first paragraph of the Results (**page 8, lines 121–127**) and further discussed this in the Discussion (**page 21–22, lines 352–364**).

3. A clear description of all the panel of DS isogenic iPSC cell lines should be made available in a table. The number of passages from the initial Trisomic cells should be mentioned for all subclones.

Response: As suggested, we have incorporated this into **Supplementary Table 1**.

4. In addition, it should be clearly indicated where all the IPS cells are made available to the scientific community.

Response: We apologize for our unclear description. We have not yet deposited these cell lines in a cell bank, but they will be made available (**page 40, lines 815–816**).

5. In the figure 2 panel F, the authors reported the fold change expression of CCT8, RCAN1, DYRK1A and C21orf33 which is above the expected 1.5x fold increase usually observed in DS cells compared to disomic controls. Can the authors provide some explanation?

Response: Thank you for your insightful comment. Several studies, including ours, have demonstrated that gene expression on chromosome 21 shows a 1.5-fold increase on average, but that changes in the individual genes are highly variable. Gene expression analysis of DS cells indicated that expression of 22% of the genes proportionally increase due to a gene-dosage effect, while 7% are amplified. The other 71% of genes are compensated (EA Yahya-Graison et al. *Am J Hum Genet.* 2007). Because overexpressed genes are likely to be involved in the DS phenotype—in contrast to the compensated genes—we selected the genes that were significantly upregulated in our preliminary experiments.

6. The panel B in figure 3 is supposed to show “178 genes with positive read counts on chromosome 21” (lane 181). Unfortunately, it is not possible to read the gene name on the figure. Thus first, the figure 3B should be made available in a larger format and a table should be included in the supplementary materials with all the FC details (Finally I found it in the Sup table 1 but no reference in the text between lane 174-190). At present although I can partially share the conclusion that some genes recovered a normal expression pattern in D+ or DRemov condition, but certainly not all of them... additional question should be raised here.

Response: We apologize for the confusion. We have accordingly enlarged panel B in Figure 3 and cited Supplementary Table 3 (**page 11, lines 173**). Although our results indicated that some genes recovered their upregulated expression pattern under the D^{remov} condition, most genes maintained their suppressed condition even after Dox removal. This persistent suppression of *XIST* was reported in previous studies

using human *XIST* (Jiang et al. 2013 *Nature*, Brown et al.1994, *Nature*). On the other hand, various reactions of genes to *XIST* were also reported, indicating that 3–7% of mouse and 12–20% of human genes on the inactivated X chromosome escape XCI. Whether this mechanism of ‘gene escape’ is same as that observed during expression recovery in D^{remov} , genomic or epigenetic mechanisms may contribute to this phenomenon. We have discussed this in the Discussion (page 22–23, lines 371–389).

7. Why the fold change of *DYRK1A* is shown on fig 3f and 4f? are the data for D- and D+ the same? If yes remove from 3F.

Response: Although these two figures represent independent experiments, we removed the figure in 2f as suggested.

8. Lane 235: the authors stated “*DYRK1A* was identified as the most potent candidate gene in the PCA list (Supplementary Table 2)”. In this table *DYRK1A* is ranked 7... thus Can the authors explain why this is the most potent candidate in the PCS list and which criteria was/were used?

Response: We apologize for our unclear description. First, 178 genes with positive expression on chromosome 21 were listed, after which 24 genes located in the 4-Mb region were ranked in order of the PCA component 1 values. We speculated that responsible genes for accelerated APC proliferation are located within this region, and because *DYRK1A* was ranked in the top position among the 24 genes (also in the 7th position among the 178 genes), it was determined as the most potent candidate. We have further modified Supplementary Table 3 and moved it to the main text as Table 1. Moreover, the gene selection method was described in detail (page 13–14, lines 211–220).

9. Figure 5e it would be better to represent P-STAT3/STAT3 and then STAT3/b-actin.

Response: **Figure 5e** was rearranged as suggested.

10. Can the authors explained or proposed a mechanism for the observation reported

lane 181: "Similarly, *DYRK1A* expression was not altered by chromosome

silencing in $\Delta M1\Delta M2$ - or $\Delta M2\Delta P$ -*DY*^{+/m/m} APCs (Supplementary Fig. 12b)."

Response: We apologize for our unclear description. *DYRK1A* was already deleted in both $\Delta M1\Delta M2$ - and $\Delta M2\Delta P$ -*DY*^{+/m/m} lines, and *DYRK1A* mRNA was not expressed from the M2 allele. Because *XIST* cDNA was inserted into the M2 chromosome, *DYRK1A* expression levels should not be affected by Dox treatment in these cell lines.

11. In the experiment to demonstrate the role of *DYRK1A*, the expression level of the gene in *XIST*-TG21 iPS showed a 2x increase. When *DYRK1A* is targeted, DeltaM1, DeltaM2 or Delta P, the level was still around a 1.5x increase. Can the author propose some explanation? Are the control disomic cells a good control? Maybe it can explain the observation reported lane 266-269.

Response: Thank you for this insightful comment. As pointed out, expression levels of *DYRK1A* in both D+ APCs and $\Delta M1$ -, $\Delta M2$ -, or ΔP -*DY*^{+/+m} APC lines remained ~1.4-fold higher than those in cDi21 APCs. We confirmed that *DYRK1A* expression in cDi21 iPSCs is equal to that in healthy control iPSCs in a previous study (Omori et al. 2017 *Sci Rep*); thus, *DYRK1A* upregulation may have resulted from the dosage effects of other genes, which were trisomic in *DY*^{+/+m} APC lines or escaped from X-inactivation in *XIST*-Tri21 APC. We have accordingly discussed this in the Discussion (**page 22, line 364–370**).

Furthermore, proliferation rates of the *DY*^{+/+m} APC line, which contains two copies of the normal *DYRK1A* allele in the Tri21 line, were higher than those of the cDi21 line. Moreover, APC proliferation was attenuated by Dox treatment in the $\Delta M2$ -*DY*^{+/+m}, $\Delta M1\Delta M2$ -, and $\Delta M2\Delta P$ -*DY*^{+/m/m} APC lines, in which *DYRK1A* expression was unaltered by Dox treatment. These results suggest that there are other critical factors on chromosome 21 involved in APC proliferation. We have included this information in the

Results (page 17–18, line 288–296).

12. Do the authors have a possibility to detect DYRK1A protein level and to measure DYRK1A kinase activity in their mutant cells and when they treated with FINDY? This will be more direct compare to looking at expression level.

Response: We measured protein and p-STAT3 levels in FINDY-treated Tri21 APCs (Supplementary Figure 13a and c).

13. Overall, all the results should be presented with individual data points rather than histogram to allow a clear view of the variability and of the number of points used in each experiment.

Response: We agree with this important suggestion and have modified our figures accordingly to present individual data points and include color-coded histograms according to the genotypes in the same figure to improve clarity.

Miscellaneous:

1. Lane 124: continuously to continuously

Response: We apologize for our mistake. This has been revised.

2. The sex of the IPS should be clearly defined (male or female)

Response: We have described that the cell line was generated from a male patient with DS in the revised Methods (page 28, lines 482–483).

=====
Reviewer #3.

The manuscript is beautifully written and the summary of the experimental setting is

clearly depicted in the first figure, which facilitates the reading of the results. With few exceptions, experiments are well controlled.

Response: Thank you very much for your positive review of our paper. We hope that the revised manuscript is now acceptable for publication in *Communications Biology*.

Comments:

1. The experiments indicate that the overexpression of DYRK1A increases APC proliferation. However, evidences obtained in animal models and cell cultures have shown that DYRK1A overexpression reduces the proliferation of NPCs. Did the authors perform proliferation assays in the different NPGs lines before culture them in the "astrocyte medium"? As mention in the Discussion, the attenuated proliferation of progenitors overexpressing DYRK1A has been attributed to alterations in Cyclin D1, p27 and p21 levels. It could be possible that the effect of DYRK1A on the stability of these cell cycle regulators is cell context-specific. Expression data of these proteins in both Tri21-NPCs and their derivative APCs and EdU incorporation data in Tri21-NPCs treated with the DYRK1A inhibitor FINDY should be easy to generate and could clarify this apparent contradiction.
2. The results in the EMBO Reports paper from Kurabayashi et al. show that the overexpression of DYRK1A increases the astroglial potential of brain NPCs during development. The authors could add a paragraph in the Discussion explaining the implications of their results in astroglialogenesis.

Response: Thank you for raising this important issue. We accordingly performed proliferation assays with NPCs and evaluated the expression levels of Cyclin D1, p21^{CIP1}, and p27 in NPC and APC lines. We found that Tri21 NPCs showed impaired proliferation and lower levels of Cyclin D1, as previously reported in *Dyrk1A* transgenic mice (**Supplementary Fig. 14**). Notably, Cyclin D1 levels were similarly reduced, and those of p21^{CIP1} were increased in Tri21 APCs, in contrast to its opposite proliferative phenotype (**Supplementary Fig. 15**). Moreover, neither copy number reductions (**Supplementary Fig. 11b**) nor inhibitor treatment (**Supplementary Fig. 13b**) of *DYRK1A* altered these protein levels in APCs, indicating that *DYRK1A* regulates cell

proliferation through different mechanisms in APCs and NPCs. Furthermore, Ser727-phosphorylated STAT3 levels were significantly decreased in not only DY^{+/+/m} and DY^{+/m/m} APCs but also in FINDY-treated APCs and NPCs (**Fig. 5p, Supplementary Fig. 13c, 16, 17**). We have included this information in the Results (**page 16–17, lines 272–287**) and Discussion (**page 25–26, lines 423–436**).

3. In Fig. 2b, the authors show immunostainings for 4 astroglial markers in XIST-Tri21 APCs cultures. It is possible that the percentage of cells expressing each of these markers varies depending on the differentiation stage of the culture, which could be influenced by the overexpression of DYRK1A and/or other Chr.21 genes. Did the authors check if the percentages of cells expressing these markers are maintained among the conditions analysed (D-, D+ and Dremov)? This information could be relevant for the interpretation of the results and should be included in the manuscript.

Response: Thank you for your insightful comment. We agree, and have thus evaluated the percentages of cells expressing GFAP, S100 β , CD44, and Vimentin in D-, D+, and D^{remov} lines, and confirmed that these astrocytic markers were stably expressed throughout these treatments. We added this in the Results and Supplementary Table 2 (**page 10, lines 153–154**).

4. *PIGP* encodes a protein of the glycosylphosphatidylinositol-anchor biosynthesis pathway with no reported functions on cell proliferation. The proliferation defects in the gain- and loss-of function experiments presented in Fig. 6 are moderate. The authors should perform additional experiments to support the dosage-dependent effect of *PIGP* on APC proliferation (i.e. same type of experiments in an additional cell line, cDI21 iPSCs for example).

Response: Thank you for this important suggestion. We performed additional proliferation assays to confirm the effect of *PIGP*, and found that co-overexpression of *PIGP* and *DSCR3* significantly increased APC proliferation, despite that single overexpression of *DSCR3* had no effect on proliferation (**Supplementary Fig 18 and 19**). Furthermore, *PIGP* overexpression accelerated the proliferation of NPCs (**Supplementary Fig 20**). These findings have been added to the Results (**page 18–19**,

lines 299–314).

Methodology:

- The number of passages of the different cell lines used in each experimental procedure should be indicated in Methods.
- The numbers of generated cell lines are in Fig. 1 (legend). The number of independent cell lines used in each experiment should also be clearly state in Methods and figure legends.

Response: As suggested by the reviewer, we have included the number of generated cell lines and the passage in **Supplementary Table 1**. We have added the number of independent cell lines used in each experiment to figure legends as necessary.

- The same for the statistical test used to compare different sets of data.

Response: We apologize for the oversight and have indicated the statistical tests used in the relevant figure legends.

Minor comments:

The sentence “In the DS brain, astrocytes may act as a primary effector in DS pathophysiology” (page 26) is an overstatement.

Response: We apologize for our incorrect description and have accordingly revised it to “In the DS brain, astrocytes may act as an important modulator in DS pathophysiology” (page 28, lines 472–473).

REVIEWERS' COMMENTS:

Reviewer #1 (Remarks to the Author):

The revised manuscript of Kawatani et al. has substantially improved. The authors have addressed all the points raised by this reviewer except one: Additional data on astrocyte morphology would be interesting to present since changes have been reported in Down syndrome brains. Can the authors present these data and run the analysis from their ICC?

Reviewer #2 (Remarks to the Author):

The Authors replied to all my comments in revised manuscript

Reviewer #3 (Remarks to the Author):

The revised manuscript by Kawatani et al. and rebuttal letter have satisfactorily addressed most of my comments and concerns. I believe this is a significantly improved version.

Still there are some minor issues and errors that need to be addressed.

- Change "DSCR" for "DSCR3" (lines 301 and 837).
- Most methods are in supplementary data. I suggest to add a note in Methods indicating the information included in Supplementary data.
- For consistency, reference 56 (line 424) should be changed for the original/s paper/s showing that "loss of DYRK1A function triggers proliferation".
- According to supplementary Table 1, experiments in Fig. 6 were done with APCs differentiated from a single line of transduced NPCs. To be consistent and avoid confusions I suggest to change in the legend "n=3 experiments per cell line" to "n=3 experiments per condition". The same in Supplementary Figs. 13, 17, 18, 19 and 20. Following the same rationale, I suggest to eliminate the sentence "Each of the data was obtained from two lines (a-c)" in the legend to Supplementary Fig 14.
- Supplementary Figs 13 and 17. Add the concentration of FINDY used in the figure or the legend.
- There is a mistake in Supplementary Table 2; "GFAP-, S100b-, CD44-".

Point-by-point responses to the Reviewers' comments

We are grateful to the editor and reviewers for their constructive comments and insightful suggestions that have helped us to improve our manuscript. As indicated in the responses below, we have taken all comments into consideration for the revision of our manuscript. For clarity, changes are highlighted in red in the revised manuscript.

=====

Reviewer #1.

The revised manuscript of Kawatani et al. has substantially improved. The authors have addressed all the points raised by this reviewer except one: Additional data on astrocyte morphology would be interesting to present since changes have been reported in Down syndrome brains. Can the authors present these data and run the analysis from their ICC?

Response: As suggested by this reviewer, Zdaniuk showed that astrocytes in DS brain were morphologically more mature than in controls of corresponding age. We agree with this reviewer's comments and believed that it would be important to examine the morphological changes in our cell lines. In our differentiation protocol, however, nearly all differentiated cells were positive for vimentin, indicating that these are astrocyte precursor cells. Because these cells were developmentally homogenous and restricted to immature stage, we found no mature astrocytes in these cells. There were no morphological differences in our immature APCs between DS and controls, which was consistent with the description in the paper, "there were no differences in vimentin staining in DS compared with controls".

=====

Reviewer #2.

The Authors replied to all my comments in revised manuscript.

Response: Thank you for your positive assessment. We hope that the revised manuscript is now acceptable for publication in *Communications Biology*.

=====

Reviewer #3.

The revised manuscript by Kawatani et al. and rebuttal letter have satisfactorily addressed most of my comments and concerns. I believe this is a significant improved version. Still there are some minor issues and errors that need to be addressed.

Comments:

1. Change “DSCR” for “DSCR3” (lines 301 and 837).

Response: We apologize for our mistake. This has been revised.

2. Most methods are in supplementary data. I suggest to add a note in Methods indicating the information included in Supplementary data.

Response: Thank you for your comment. We followed your comment and editor’s suggestion and moved most part of the methods to the manuscript.

3. For consistency, reference 56 (line 424) should be changed for the original/s paper/s showing that “loss of DYRK1A function triggers proliferation”.

Response: We changed this reference to original papers (Soppa et al. 2014, Hammerle et al. 2011).

4. According to supplementary Table 1, experiments in Fig. 6 were done with APCs differentiated from a single line of transduced NPCs. To be consistent and avoid confusions I suggest to change in the legend “n=3 experiments per cell line” to “n=3 experiments per condition”. The same in Supplementary Figs. 13, 17, 18, 19 and 20. Following the same rational, I suggest to eliminate the sentence “Each of the data was obtained from two lines (a–c)” in the legend to Supplementary Fig 14.

Response: Thank you for your comment. We agreed and changed our description in Fig. 6, and Supplementary Figs 13, 14, 17, 18, 19 and 20.

5. Supplementary Figs 13 and 17. Add the concentration of FINDY used in the

figure or the legend.

Response: We added the value in the legend.

6. There is a mistake in Supplementary Table 2; “GFAP-, S100b-, CD44-“.

Response: We apologize for our ambiguous description. We changed this phrase to “the average percentage of cells positive for GFAP, S100b, CD44, and vimentin in...”.